# IDH status dictates oHSV mediated metabolic reprogramming affecting anti-tumor immunity

Upasana Sahu [1,2] ✉, Matthew P. Mullarkey[3], Sara A. Murphy[1,2,4], Joshua C. Anderson[5], Vasanta Putluri[6], Abu Hena Mostafa Kamal [6,7], Jun Hyoung Park[8], Tae Jin Lee [9], Alexander L. Ling[10], Benny A. Kaipparettu [8], Ashok Sharma [9], Nagireddy Putluri [6,7], Pamela L. Wenzel [11], Christopher D. Willey [5], E. Antonio Chiocca [10], James M. Markert [12] & Balveen Kaur [1,2] ✉

Identification of isocitrate dehydrogenase (IDH) mutations has uncovered the crucial role of metabolism in gliomagenesis. Oncolytic herpes virus (oHSV) initiates direct tumor debulking by tumor lysis and activates anti-tumor immunity, however, little is known about the role of glioma metabolism in determining oHSV efficacy. Here we identify that oHSV rewires central carbon metabolism increasing glucose utilization towards oxidative phosphorylation and shuttling glutamine towards reductive carboxylation in IDH wildtype glioma. The switch in metabolism results in increased lipid synthesis and cellular ROS. PKC induces ACSL4 in oHSV treated cells leading to lipid peroxidation and ferroptosis. Ferroptosis is critical to launch an anti-tumor immune response which is important for viral efficacy. Mutant IDH (IDHR132H) gliomas are incapable of reductive carboxylation and hence ferroptosis. Pharmacological blockade of IDHR132H induces ferroptosis and anti-tumor immunity. This study provides a rationale to use an IDHR132H inhibitor to treat high grade IDH-mutant glioma patients undergoing oHSV treatment.

Oncolytic herpes simplex type 1 (oHSV) is an immune therapy that employs attenuated viruses to lyse tumor cells and engage immunity to destroy cancer[1]. Imlygic (for melanoma) and Delyatect (for high-grade glioma) are the two oHSVs that are currently approved for the treatment of cancer[2]. While multiple other viruses continue to be evaluated in clinical trials for safety and evidence of efficacy, it is critical to understand how these viruses affect the tumor and its microenvironment to develop more effective strategies to treat glioma.

The most significant discovery in the last decade that has impacted neuro-oncology has perhaps been the identification of a gain

[1]Department of Pathology, Medical College of Georgia at Augusta University, Augusta, GA, USA. [2]Georgia Cancer Center at Augusta University, Augusta, GA, USA. [3]Department of Neurosurgery, McGovern Medical School, The University of Texas Health Science Center at Houston, Houston, TX, USA. [4]University of Texas MD Anderson Cancer Center, UTHealth Houston Graduate School of Biomedical Sciences, Houston, TX, USA. [5]Department of Radiation Oncology, Marnix E. Heersink School of Medicine, University of Alabama at Birmingham, Birmingham, AL, USA. [6]Advanced Technology Cores, Dan L Duncan Comprehensive Cancer Center, Baylor College of Medicine, Houston, TX, USA. [7]Department of Molecular and Cellular Biology, Baylor College of Medicine, Houston, TX, USA. [8]Department of Molecular and Human Genetics, Baylor College of Medicine, Houston, TX, USA. [9]Center for Biotechnology and Genomic Medicine, Medical College of Georgia at Augusta University, Augusta, GA, USA. [10]Department of Neurosurgery, Brigham and Women's Hospital, Boston, MA, USA. [11]Department of Integrative Biology & Pharmacology, McGovern Medical School, The University of Texas Health Science Center at Houston, Houston, TX, USA. [12]Department of Neurosurgery, Marnix E. Heersink School of Medicine, The University of Alabama at Birmingham, Birmingham, AL, USA. ✉ e-mail: usahu@augusta.edu; bkaur@augusta.edu

of function mutation in the enzyme isocitrate dehydrogenase (IDH)[3,4]. IDH is a metabolic enzyme that catalyzes the decarboxylation of iso-citrate to α-ketoglutarate (αKG) to maintain balanced levels of iso-citrate and αKG. Frequently identified oncogenic *IDH* mutations in glioma are a gain of function that results in the production of 2-hydroxyglutarate (2-HG), an oncometabolite that disrupts cellular metabolism and also affects DNA methylation[3,5]. Changes in the IDH status of glioma clearly affect tumor cell metabolism that can affect both cancer cell-intrinsic bioenergetic states and also have far-reaching effects on cellular signaling, stromal cell reprogramming, and patient prognosis[6]. Thus, it is highly important to study the impact of changes in tumor bioenergetics on the therapeutic index of novel therapies.

While oHSV therapy is being used to treat GBM patients in Japan and is in multiple clinical trials for patients in the USA and Europe, its effect on the bioenergetic state of cancer cells is not well studied. Here, we investigate the impact of oHSV treatment on cellular bioenergetic states and the effects of these changes on immune cell reprogram-ming. Transcript profiling of bulk RNA sequencing of pre- and post-treatment biopsies of brain tumor patients treated with G207 and CAN-3110, two different oHSVs, reveals enrichment in gene sets that predict changes in cellular metabolic processes. Our results highlight the role played by IDHR132H in curbing the immune benefit of vir-otherapy and demonstrate that the inhibition of IDHR132H function can improve the efficacy of oHSV therapy.

## Results

### oHSV therapy-induced changes in glucose flux in patients and cell models

To analyze if oHSV treatment-induced changes in cellular metabolism we analyzed transcriptome profiles of patients before and after treat-ment. Gene set enrichment analysis (GSEA) of sequencing data from NCT00028158 which examined the effect of G207 on patients diag-nosed with recurrent GBM (rGBM)[7] (three pretreatment and three post-treatment biopsies) uncovered a significant enrichment in path-ways related to citric acid cycle, oxidative phosphorylation, and glu-tamate metabolism (Figs. 1a–d and S1). Analysis of transcriptomic data from another recent trial NCT03152318, evaluating oHSV CAN-3110 in high-grade brain tumor patients also revealed enrichment of GSEA pathways related to oxidative phosphorylation, mitochondrial respiration, and ATP synthesis in HSV seropositive patients but not in HSV seronegative patients (Fig. 1e–k)[8]. It is interesting to note that in this trial it was HSV seropositivity that correlated with predicted response in patients[9]. Collectively, this suggested that oHSV therapy not only changed cellular bioenergetics but that it might also affect response to treatment.

To further investigate this we incubated glioma cells treated with two different oHSVs (HSVQ (Q) or HSV-P10 (P)) with stable isotope-labeled U-13C glucose (Figs. 2 and S2)[10,11]. Increased Glucose-6-Phos-phate/Fructose-6-Phosphate (G6P/F6P) and 3-Phosphoglycerate-2-Phosphoglycerate (3PG-2PG), with no increase in metabolites of pen-tose phosphate pathway, indicated oHSV treatment-induced increased glucose utilization and increased flux into the glycolytic pathway (Figs. 2a–c and S2a–c). Increased pyruvate kinase activity (an enzyme that catalyzes the conversion of phosphoenolpyruvate to pyruvate) also corroborated with increased flux into the glycolytic pathway (Fig. 2d). This was further accompanied by increased pyruvate dehy-drogenase activity and its catalytic product acetyl CoA in glioma cells treated with oHSV (Figs. 2e, f and S2d). Metabolic flux analysis using U-13C glucose in GBM12 cells revealed an overall increase in U-13C glucose incorporation (6 h) into the TCA cycle in P-infected glioma cells compared to uninfected control cells (*n* = 6) resulting in increased labeled citrate and succinate (Figs. 2g, h and S2e). This suggests that oHSV treatment leads to increased glucose utilization by glioma cells and metabolizes excess glucose into the TCA cycle via acetyl CoA.

Increased malate with three labeled carbons (M + 3) also showed increased pyruvate carboxylation to malate into the TCA via pyruvate/malate cycle (Fig. 2i). Despite the increased flux in glucose utilization, there was no increase in lactate (Fig. S2f). Collectively, this data sug-gested that oHSV treatment increased glucose uptake by tumor cells that were preferentially guided into the TCA cycle.

### oHSV-infected GBM cell mitochondria exhibit increased mito-chondria activity

To evaluate if oHSV treatment affected the mitochondrial oxidative phosphorylation pathway we performed Seahorse analysis which simultaneously measures glycolytic activity and rate of mitochondrial respiration in GBM12 and GBM28 glioma cells treated with HSVQ or HSV-P10 respectively. Mitochondria stress assay by sequentially blocking ATP synthase activity to analyze mitochondrial ATP produc-tion, ATP-independent maximal mitochondrial activity, and non-mitochondrial respiration was performed as described[12]. oHSV treat-ment induced a significant increase in mitochondrial respiration in both GBM12 and GBM28 treated cells, along with increased basal respiration and ATP production (Figs. 3a–d and S3a, b). Treatment induced an increase in maximum respiratory capacity in oHSV-infected GBM cells along with no change in proton leak (Fig. S3c, d) and unal-tered non-mitochondrial respiration (Fig. S3e) indicating that oHSV infection increases mitochondrial capacity and that the infected cells primarily rely on mitochondria to meet their oxygen/energy demands. Despite increased glucose utilization and mitochondrial activity, we did not see a significant increase in extracellular acidification rate (ECAR) after oP10 treatment (Fig. 3e). Interestingly, while HSVQ treatment of both GBM12 and GBM28 cells showed increased oxygen consumption, there was also a slight increase in ECAR suggesting that virus-induced glucose flux into mitochondrial activity was more pro-nounced in oP10 treated cells (Fig. S3f, g). Consistent with increased oxidative phosphorylation and TCA cycle activity, mito-tracker stain-ing also demonstrated that both GBM12 and GBM28 cells had more mitochondrial ROS (Fig. 3f, g). Mitochondrial and genomic DNA PCR showed increased mitochondria content and evaluation of activity of respiratory chain enzymes also showed increased mitochondrial respiratory chain complex activities (Fig. 3h, i). Collectively, this indi-cated that treatment with oHSV induced changes in mitochondrial activity of the glioma cells.

### oHSV induces high metabolic flux through reductive carbox-ylation in GBM cells

Glutamine is the most abundant non-essential amino acid in the human body and a key respiratory substrate in tumor cells in pro-viding energy and carbon for growth by anaplerotic replenishment of TCA cycle intermediates to feed biosynthetic pathways[13,14]. GBM cells are known to use glutamine as a carbon source for lipid synthesis and to satisfy nitrogen requirements[13,15]. Hence, we used U-13C labeled glutamine to examine its metabolic fate in oHSV-treated cells (Fig. 4a). Overall, there was a slight but significant reduction in labeled glutamine and glutamate levels in treated cells which could be indicative of increased flux in the pathway (Figs. 4b, c and S4a, b). Glutamate generated α-ketoglutarate (α-KG), can be either (i) oxi-dized in the TCA cycle (which would result into M + 4 intermediates in TCA) or can be carboxylated into M + 5 isocitrate and citrate (purple arrows) by reductive carboxylation (Fig. 4a)[16]. M + 5 citrate was the most abundant species of labeled citrate in oHSV-infected GBM cells (12%) compared to 6% in uninfected control cells (Figs. 4d and S4c). oHSV-infected GBM cells also showed a 2-fold increased ratio of M + 5 citrate to M + 5 α-KG indicating that a majority of the glutamine utilized upon infection was converted to citrate by reductive carboxylation and not through the TCA cycle (Figs. 4e and S4d). TCA cycle intermediates (malic acid, fumaric acid, succinic acid and oxaloacetic acid) derived from [U-13C] labeled

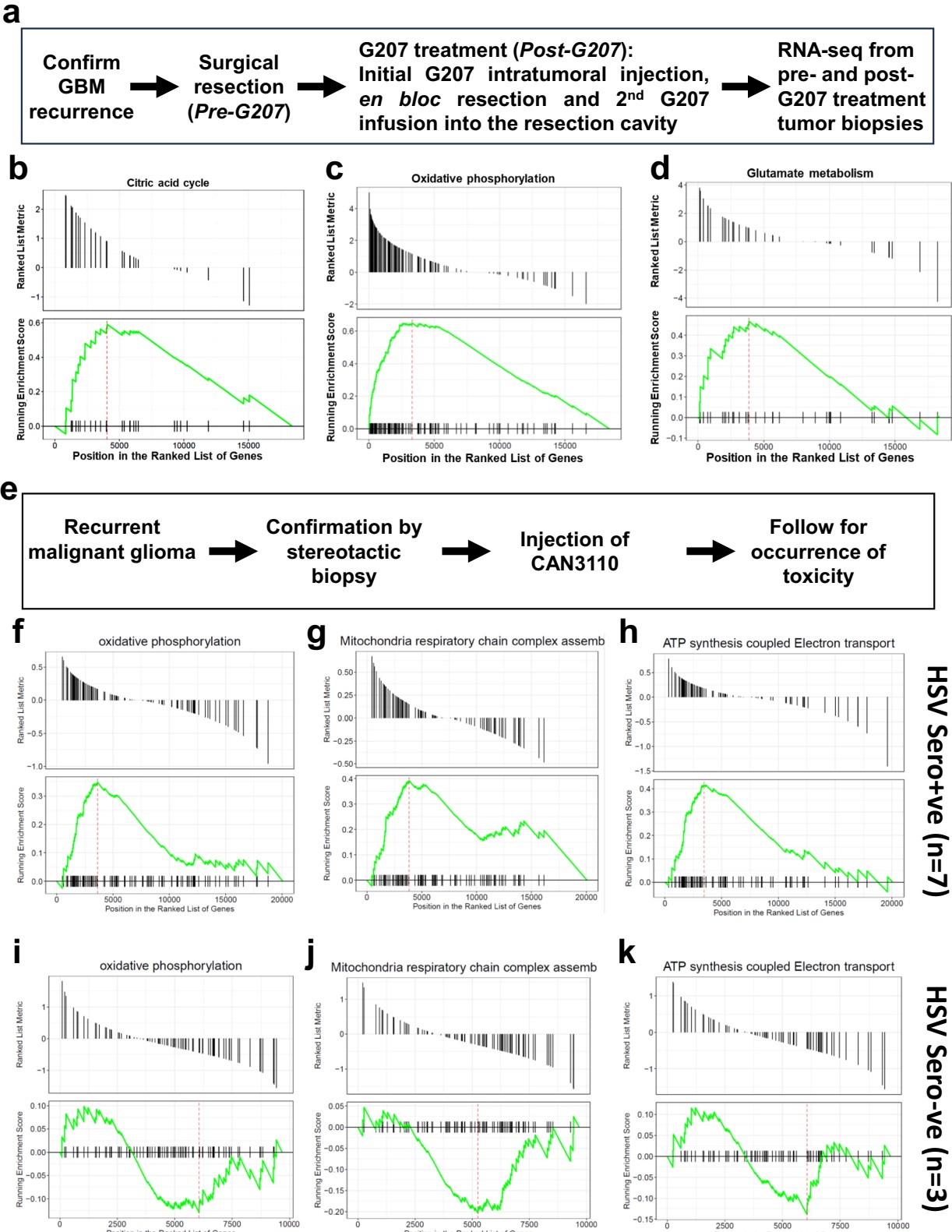

**Fig. 1 | Effects of oHSV treatment on mitochondria dynamics in glioma patients. a** Overview of the NCT00028158 clinical trial design and sample collection[7]. GSEA of RNA sequencing data from GBM patients (*n* = 3) before and after oHSV G207 treatment depicting differentially expressed genes involved in the citric acid cycle (**b**), oxidative phosphorylation (**c**) and glutamate metabolism (**d**) (post- vs pre-oHSV G207 treatment). **e** Overview of the NCT03152318 clinical trial design[8]. **f–k** GSEA of RNA sequencing data from recurrent glioma patients before and after oHSV CAN-3110 treatment in patients positive for HSV serotype (*n* = 7 for sero +ve) or negative (*n* = 3 for sero −ve). Positive enrichment of genes involved mitochondrial oxidative phosphorylation (**f, i**), respiratory chain complex assembly (**g, j**) and ATP synthesis coupled electron transport (**h, k**) in CAN-3110 treated patients positive for HSV serotype (**f–h**) as opposed to the HSV seronegative patients (**i–k**).

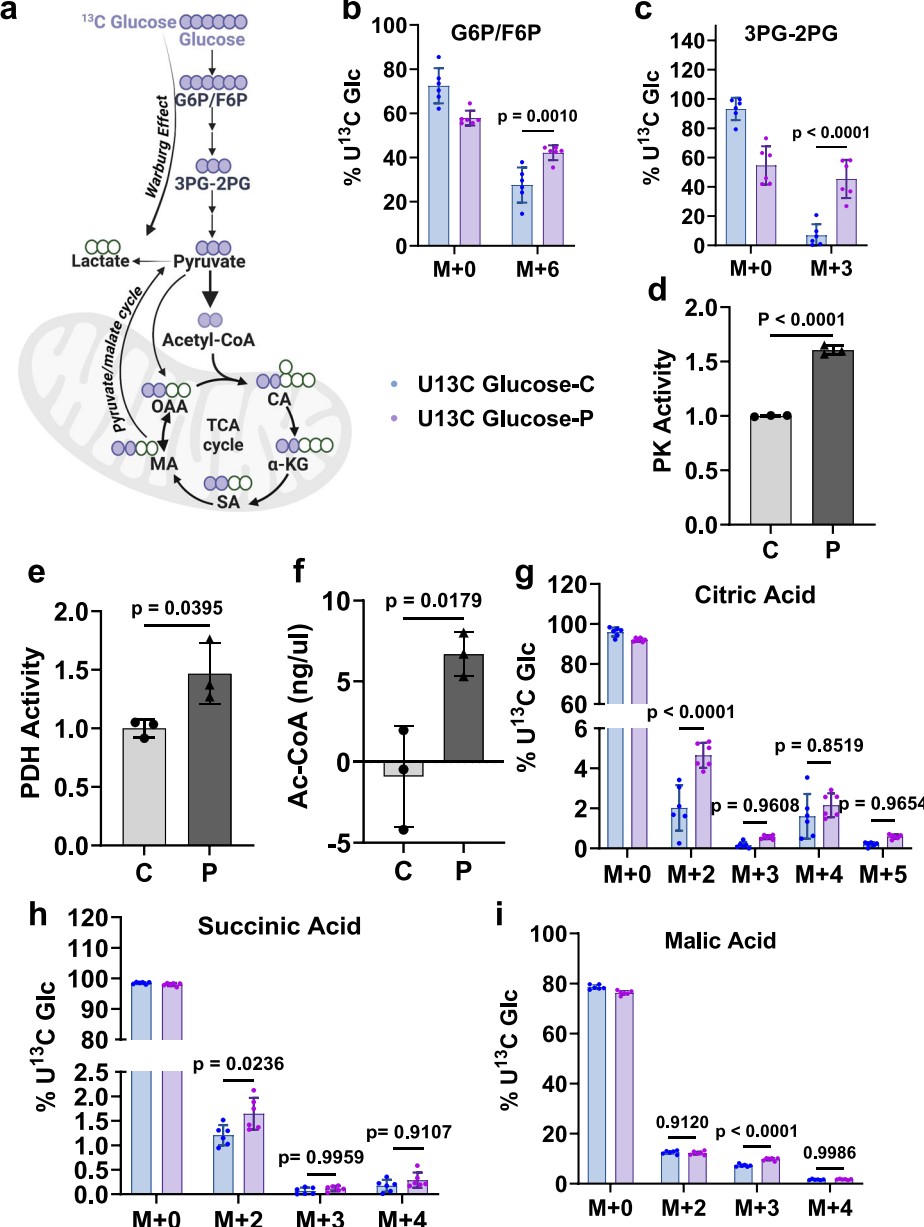

**Fig. 2 | Glucose contribution to TCA cycle metabolites in oHSV-infected cells.**
Effect of oHSV infection on central carbon metabolism. **a** Schematic of glucose
metabolites in glycolysis and TCA cycle. Filled purple circles indicate the labeled
carbons from [U-13C] glucose in the first round in TCA cycle. [Created in BioRender.
Sahu, U. (2025) https://BioRender.com/s86n709]. GBM12 cells infected with oP10
(P) at MOI = 0.01 for 12 h were cultured in glucose-free media for 4 h followed by
supplementation with U-13C glucose and 2% dialyzed FBS for 6 h were analyzed by
LC/MS (*n* = 6/group) for the indicated metabolites (M + *n*, where *n* refers to the
number of labeled carbon atoms in the metabolite). Total percentage of the
indicated metabolite labeled with $^{13}$C for Glucose-6-Phosphate/Fructose-6-Phos-
phate (**b**) and 3-Phosphoglycerate/2-Phosphoglycerate (**c**) (*n* = 6). **d** Pyruvate kinase
(PK) activity in nmol/min/mL (*n* = 3) and **e** Pyruvate dehydrogenase (PDH) activity
in nmol/min/mL in lysates from control (C) or P-treated GBM12 cells (*n* = 3).
**f** Concentration of total Acetyl-CoA (ng/μL) in control and oHSV-infected GBM12
cells (*n* = 3). Percentage of Citric acid (**g**), Succinic acid (**h**), and Malic acid (**i**) labeled
with $^{13}$C from U-13C glucose (*n* = 6). Data are shown as mean ± S.D. (*n* ≥ 3 indepen-
dent replicates). (Two-way ANOVA, unpaired two-tailed Student's *t*-test). MOI =
Multiplicity of infection. Source data are provided as a Source Data file.

glutamine would be labeled on M + 4 by forward cycling in the TCA
cycle (via α-KG) or M + 3 if derived from citrate (Fig. 4a). There was no
increase in M + 3 or M + 4 TCA cycle intermediates (Fig. S4e–h). This
suggests that the increased M + 5 citrate likely contributed to fatty
acid synthesis and/or protein acetylation[16]. oHSV-treated cells had an
overall increase in acetyl CoA levels (Fig. 2f), increased lipid synthesis
as seen by increased Nile red staining (Fig. 4f, g), and lipid droplets in
transmission electron microscopy images (Fig. 4h, i) and increased
levels of relative fatty acid content (Figs. 4j, k and S4j, k). Consistent
with this, GSEA analysis of RNA-seq data from tumor biopsies of

rGBM patients before and after oHSV G207 treatment also showed
significant enrichment of genes involved in lipid metabolism with
oHSV treatment (Fig. S4l). Taken together, our data indicates that
oHSV treatment of glioma cells resulted in increased glutamine uti-
lization towards reductive carboxylation leading to increased lipid
synthesis.

**oHSV treatment induces cell death with hallmarks of ferroptosis**
Increased flux of glucose into the TCA cycle along with increased
oxygen consumption would predict increased ROS. Increased

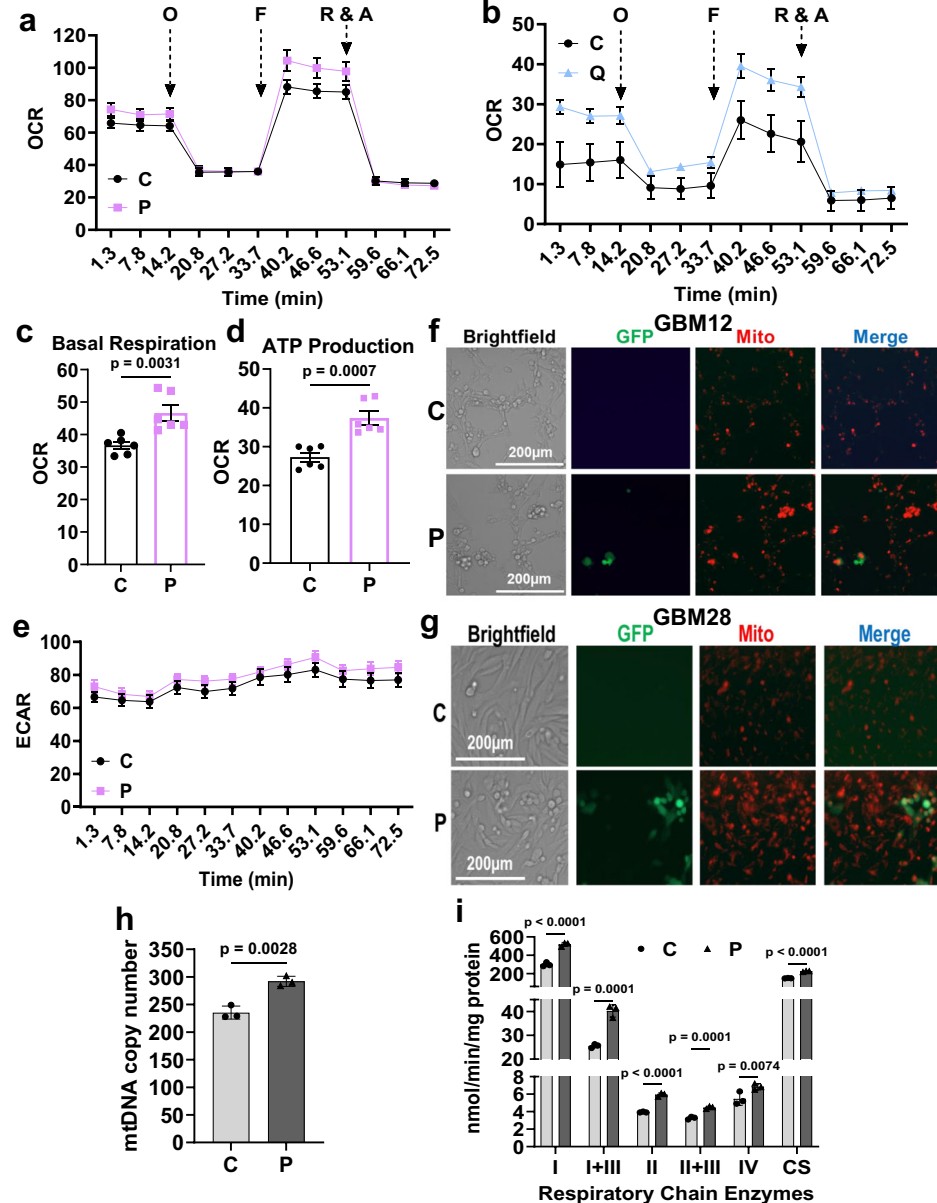

**Fig. 3 | Increased mitochondria activity in oHSV-infected GBM cells.**
**a–e** Seahorse Bioscience XF96 extracellular flux analyzer was used to measure oxygen consumption rate (OCR) in pMoles/min, indicative of OXPHOS in uninfected and oHSV-infected GBM12 and GBM28 cells 24 hpi (MOI = 0.01). OCR of control (C) or oP10 (P) treated GBM12 cells ($n = 6$) (**a**) and C and HSVQ (Q) treated GBM28 ($n = 3$) (**b**) cells in pmol/min/8 × 10⁴ in real-time under basal conditions and in response to mitochondrial inhibitors (oligomycin (O); FCCP (F); rotenone (R) and antimycin (A)). Quantification of basal respiration (**c**) and ATP production (**d**) represented as OCR (pmol/min/8 × 10⁴ cells) ($n = 6$). **e** Extracellular acidification rate (ECAR) in uninfected and oHSV P-infected GBM12 cells 24 hpi ($n = 6$; Data = mean ± SEM). Mitotracker stain showing actively respiring mitochondria fraction in uninfected and P-infected GBM12 (**f**) and GBM28 (**g**) cells 24 hpi. Scale bar = 200 μm. **h** Mitochondria DNA (mtDNA) copy number/haploid genome quantified by qPCR ($n = 3$). **i** Electron transport complex activities were determined as described in the materials and methods and normalized with protein concentration. Activities of complex I, I + III, II, II + III, IV, and CS in uninfected and oHSV-infected GBM12 cells ($n = 3$). Complex I: NADH: Ferricyanide dehydrogenase; Complex I + III: NADH: cytochrome c reductase; complex II: Succinate dehydrogenase; Complex II + III: Succinate: cytochrome c reductase; Complex IV: Cytochrome c oxidase; CS: Citrate synthase. Data represents the mean value ± S.D. from at least three independent replicates. (Two-way ANOVA, Unpaired two-tailed Student's $t$-test). hpi hours post-infection. Source data are provided as a Source Data file.

glutamine flux towards reductive carboxylation would also be predicted to reduce cellular glutamate essential to produce glutathione, the quintessential cellular ROS scavenger. Thus, we checked for changes in the cellular and mitochondrial ROS along with glutathione levels. Our data revealed an increase in cellular and mitochondrial ROS with a simultaneous decrease in reduced glutathione levels in GBM cells after virotherapy (Figs. 5a–e and S10). To identify the major signaling kinases modified by oHSV treatment that dictated cellular fate, we performed a functional screen of protein kinases (Ser/Thr kinases and Tyr kinases) in primary patient-derived GBM

cells using a Pamstation12 kinomics platform. The kinome profile was measured in uninfected and oHSV-infected GBM cells. Unsupervised hierarchical clustering identified altered phosphorylated peptide signals upon oHSV infection of the GBM cells (Fig. S5a). Changes in phosphorylation of the substrate peptides (-144 substrates per array for Ser/Thr and Tyr kinases) were used to predict changes in kinase activity using BioNavigator (v6.3) and kinases were overlaid on network models using GeneGo Metacore™ and Clarivate™ (portal.genego.com). Activated Protein Kinase C (PKC) after virus treatment was identified as a well-connected node (Fig. 5f). Further, PKC target

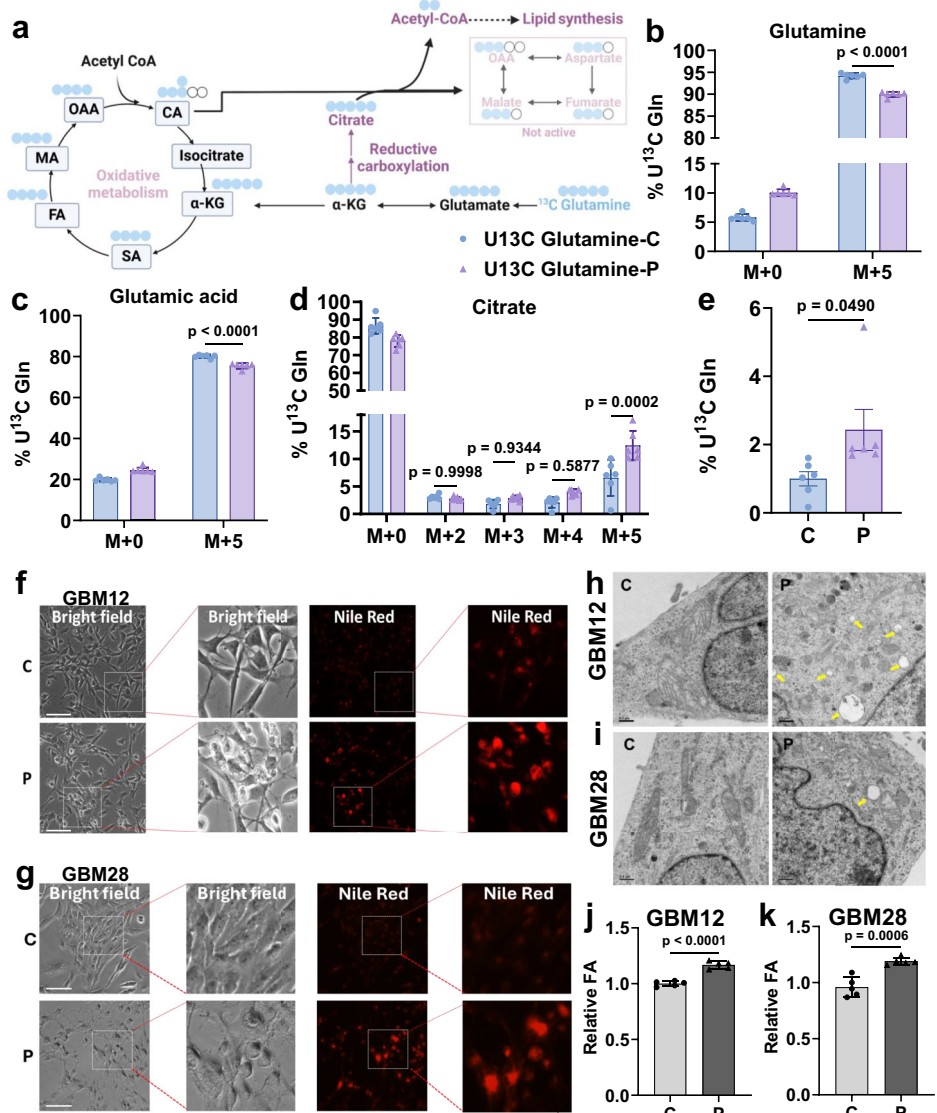

**Fig. 4 | A reductive pathway of glutamine metabolism in GBM cells following oHSV infection.** P infection results in increased metabolism of glutamine to α-Ketoglutarate which is further processed via reductive carboxylation to generate citrate. **a** Schematic depicting glutamine utilization via TCA cycle (black arrows) and reductive carboxylation of glutamine metabolism (purple arrows) with [U-13C] Glutamine. 6 h post incubation with U-13C glutamine and 2% dialyzed FBS, GBM12 cells were analyzed by LC/MS for the indicated metabolites ($n = 6$/group). The Y-axis indicates the relative % of each metabolite from the pool that is tracer-labeled by the indicated number of carbons in C- and P-treated GBM12 cells. M + $n$: where $n$ indicates the number of carbon atoms labeled with $^{13}C$ in the indicated metabolite. Blue circles are used to display the predicted fate of 13C atoms in U-13C Glutamine in one cycle. [Created in BioRender. Sahu, U. (2025) https://BioRender. com/b13v120]. Total percentage of $^{13}C$ glutamine labeled metabolites: Glutamine (**b**) and Glutamate (**c**) ($n = 6$). Levels of citrate derived from U13-C labeled gluta-mine (**d**) and fold change in the ratio of M + 5 citrate to M + 5 α-KG indicating the portion of citrate derived from α-KG via reductive carboxylation (**e**) ($n = 6$). Increased lipid droplets indicative of lipid synthesis in P-treated GBM12 (**f**, **h**) and GBM28 (**g**, **i**) cells depicted by Nile red staining, scale bar = 200 μm (**f**, **g**), and transmission electron microscopy (TEM), scale bar = 0.5 μm (**h**, **i**). Relative free fatty acid in P-infected GBM12 (**j**) and GBM28 (**k**) cells relative to uninfected control cells ($n = 3$). Data are shown as mean ± S.D., $n ≥ 3$ independent replicates. (Two-way ANOVA (**b**–**d**), unpaired one-tailed Student's $t$-test (**e**), unpaired two-tailed Stu-dent's $t$-test (**j**–**k**)). C control, P oP10 infected. Source data are provided as a Source Data file.

peptides were identified as altered in cells treated with two different oHSVs (Fig. S5b, c). Increased PKC activity was further confirmed by PKC kinase activity assays and western blot analysis (Figs. 5g, h and S5d, e).

PKC plays a key role in fatty acid oxidation (one of the hallmarks of ferroptosis) via activation of acyl-CoA synthetase long-chain family member 4 (ACSL4), a key enzyme in lipid peroxidation[17]. We found that oHSV treatment induced ACSL4 transcript and protein expression along with a significant increase in lysophosphatidylcholine acyl-transferase 3 (LPCAT3) mRNA and protein which esterifies poly-unsaturated fatty acids-CoA (PUFAs-CoA) generated by ACSL4 into phospholipids, accompanied by reduced glutathione peroxidase (GPX4), a key ferroptosis marker and a critical regulator of cellular oxidative homeostasis (Figs. 5i and S5f–h)[18].

A significant increase in carnitine palmitoyl transferase 1A (CPT1A), an enzyme that is essential for fatty acid oxidation was also seen in treated cells (Figs. 5j and S5i, j)[19]. oHSV treatment also resulted in increased lipid peroxidation (LPO) and an increase in the mean fluorescence intensity (MFI) of FerroOrange (FO) positive cells (Figs. 5k–n and S10). Transmission electron microscopy of infected glioma cells also showed that oHSV-treated cells displayed hallmarks of ferroptosis that included mitochondrial fission: smaller and increased number of mitochondria (yellow arrows; yellow dashed line for one mitochondrion in control), and evidence of membrane damage

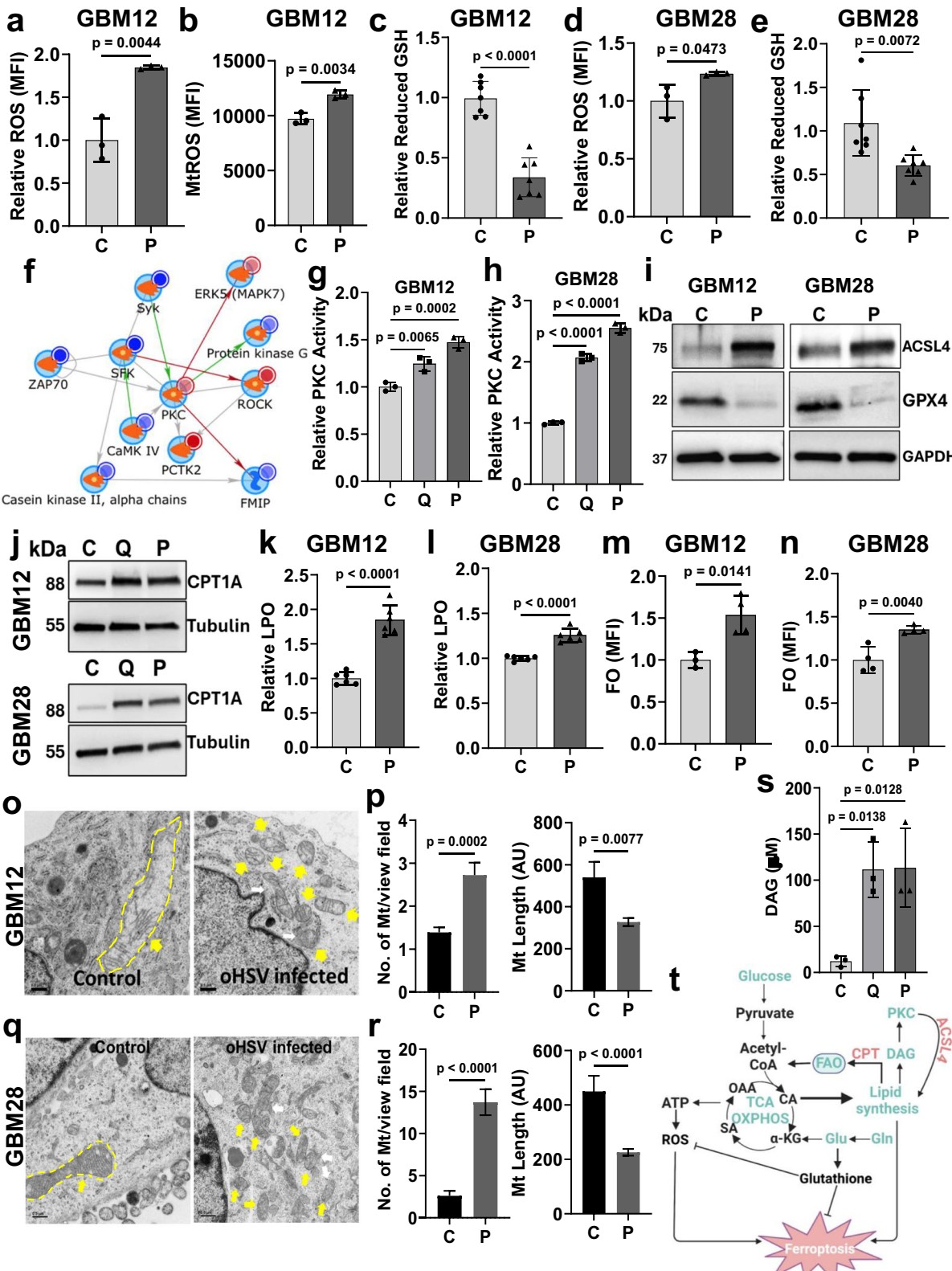

(white arrowheads) (Fig. 5o–r). Along with increased lipid synthesis, there was increased diacylglycerol (DAG) needed for PKC activation (Fig. 5s). Dot plots of gene ontology (GO) enriched pathways of the sequencing data from NCT00028158 trial which examined the effect of G207 on patients diagnosed with rGBM further showed positive enrichment of genes regulating ROS pathway and ferroptosis in patients following G207 treatment (Fig. S1e)[7]. Collectively our data

shows that virus treatment increases glucose flux into the TCA cycle which increases cellular ROS. At the same time, oHSV treatment shifts glutamine utilization towards reductive carboxylation to induce fatty acid synthesis. PKC activation induces enzymes such as ACSL4. TEM microscopy further shows smaller mitochondria going through fission with evidence of membrane damage, all considered to be hallmarks of ferroptosis. (Fig. 5t).

**Fig. 5 | Induction of ferroptotic cell death in oHSV-infected GBM cells.** Intracellular (**a**) and mitochondrial (**b**) ROS measured by flow cytometry in C- and P-treated GBM12 cells (n = 3). **c** Mean levels of reduced free Glutathione (GSH) measured 48 hpi in uninfected and infected GBM12 cells (n = 7). Intracellular ROS (**d**) (n = 3) and relative reduced GSH (**e**) (n = 7) in C- and P-infected GBM28 cells. **f–h** Lysates from uninfected and Q- or P-infected GBM12 cells (MOI = 0.01, 48 hpi) were analyzed utilizing the Tyrosine (PTK) arrays or Serine/Threonine (STK) arrays (n = 4). **f** Combined Q and P altered kinases were mapped to the direct interaction network utilizing GeneGo MetaCore™ provided by Clarivate™. Input nodes (kinases) are indicated by large blue circles; smaller circles to the top right of nodes indicate directionality (mean kinase score, MKS), red = increase, blue = decrease. Arrows between nodes indicate interactions (green positive, red inhibitory, and gray as other), and arrowhead indicates directionality. Relative PKC kinase activity of P and Q infected GBM12 cells (**g**) and GBM28 cells (**h**). **i** Western blot of ACSL4 and GPX4 in C- and P-treated GBM cells. **j** Western blot of uninfected and Q- or P-infected GBM cells for CPT1A levels. Tubulin was used as a loading control. Levels

of lipid peroxidation in uninfected and P-infected GBM12 (**k**) and GBM28 (**l**) cells (48 hpi, n = 6). Flow cytometry analysis showing relative MFI of FerroOrange (FO, 1 μM) in P-infected GBM12 (**m**) and GBM28 (**n**) cells vs C (n = 4). **o–r** Transmission electron microscopic (TEM) images of GBM12 (**o**) and GBM28 (**q**) cells uninfected or infected with P 24 hpi. Scale bar = 0.5 μm. The yellow dashed line indicates mitochondrion length in control cells. Yellow arrow = mitochondrion; White arrow = damaged outer mitochondria membrane. Quantification of TEM images in GBM12 (C: n = 21, P: n = 18 view field) (**p**) and GBM28 (C: n = 10, P: n = 7 view field) (**r**) cells. Left panel: number of mitochondria/view field, right panel: mitochondria length. **s** Levels of diacylglycerol (DAG) in uninfected and infected GBM12 cells expressed as μM (n = 3). **t** Figure summarizing oHSV-induced ferroptosis in GBM cells. [Created in BioRender. Sahu, U. (2025) https://BioRender.com/j23i805]. Data = mean ± S.D., n ≥ 3 independent replicates. (One-way ANOVA, unpaired two-tailed Student's t-test). MFI mean fluorescence intensity, hpi hours post-infection, C control, P oP10 infected. Source data are provided as a Source Data file.

## Changes in metabolism as well as PKC activity are needed for oHSV-induced lipid peroxidation-mediated tumor cell death

To evaluate the importance of glycolytic flux, reductive carboxylation, and PKC activation for oHSV-induced lipid peroxidation and immunogenic cell death, we individually blocked each of these pathways (Fig. 6a). Treatment of cells with the mitochondrial disruptor Devimistat[20] reduced the ability of the virus to induce lipid peroxidation in both GBM12 and GBM28 glioma cells (Fig. 6b, c). Treatment of both GBM12 and GBM28 cells with ROS scavenger, Ebselen[21] also reduced virus-induced lipid peroxidation (Fig. 6d, e), underscoring the importance of both treatment-induced ROS and mitochondrial activity.

To evaluate the importance of reductive carboxylation, we blocked glutamine metabolism using Telaglenastat (CB-839; NCT03875313, NCT03528642), a glutaminase inhibitor (GLSi). GLSi did not affect the ability of the virus to infect or kill tumor cells in vitro (Figs. 6f, g and S6a), but reduced lipid peroxidation (Fig. 6h, i). To interrogate the role played by PKC in virus-induced ferroptosis, we utilized Go6983, a broad-spectrum PKC inhibitor[17]. Go6983 reduced oHSV-induced lipid peroxidation and depletion of GSH in treated glioma cells (Figs. 6j, k and S6b, c). This was further confirmed by western blot analysis for lipid peroxidation product, 4-Hydroxynonenal (4-HNE), ACSL4, and GPX4, showing a reduction in ACSL4 and 4-HNE levels with an increase in GPX4 expression with PKC inhibition (Fig. 6l). Collectively these results show that modulation of glucose and glutamine flux along with PKC activation was essential for oHSV-induced ferroptosis. To evaluate the significance of ferroptosis for virotherapy we treated mice bearing 005 GSC brain tumors with oHSV and with or without GLSi to block ferroptosis. GLSi treatment reduced the therapeutic benefit of viro-immune therapy, underscoring the importance of ferroptosis in oHSV-induced anti-tumor immune benefit (Fig. 6m).

It is interesting to point out that glutamate is used by the cellular xCT transporter to bring in cystine, an essential component for the synthesis of glutathione, the cellular defense against oxidative damage. Thus, blockade of GLS activity would be predicted to increase ferroptosis and has been shown to do that in some studies[22]. Here we discovered that it reduces virus-induced ferroptosis. Consistent with our results blockade of GLS activity has also been shown to rescue peroxide-induced cell death[23]. Thus, the impact of GLS activity on ferroptosis appears to be context-dependent.

### oHSV-induced ferroptosis is important for virus-induced immune response

To evaluate the immunogenic effect of oHSV-induced ferroptosis, we measured the levels of common damage-associated molecular pattern molecules (DAMPs) such as HMGB1 and extracellular ATP (eATP), released by GBM cells undergoing ferroptosis[24]. oHSV infection

resulted in a significant increase in DAMPs release (Fig. 7a). Functionally, donor PBMC-derived dendritic cells (DCs) cultured in the presence of supernatants from uninfected or virus-infected cells were analyzed for their ability to activate T cells (Fig. S7a). Virus-infected cell supernatants not only induced HLA-DR on DCs but also led to the activation of CD8+ T cells as shown by increased count of CD8+/CD69+ T cells (Figs. 7b, c, S7b and S11). This depended on induction of ferroptosis, as treatment of virus-treated cells with ferrostatin-1 (Fer-1), an inhibitor of ferroptosis, reduced virus-induced lipid peroxidation, mitochondria damage, DAMPs release, and subsequent DC activation (Figs. 7d–g, S7c–e and S11). The therapeutic benefit of oP10 in 005 glioma-bearing mice depended on an intact immune system as this advantage was not observed in immune-deficient mice (Figs. 7h and S7f). To evaluate the effect of ferroptosis on glioma anti-tumor immunity, glioma-bearing mice were fed a cysteine-depleted diet (CMD) as described[25]. Intracranial glioma-bearing mice on a CMD diet had reduced tumor growth with improved survival that was similar to the therapeutic advantage of oP10 treatment in immune-competent but not in immune-deficient mice (Fig. 7i, j). The combination of oP10 treatment with the CMD diet did not additively increase the anti-tumor immune benefit of virotherapy (Figs. S7g and 7i, j) implying that immune therapeutic benefit from oP10 and CMD diet was driven by a similar mechanism. Additionally, treatment with either Fer-1 (Figs. 7k and S7h, i) or PKC pathway inhibitor Enzastaurin (Enza) (Figs. 7l and S7h, i) blocked therapeutic benefit of virotherapy in immune-competent animals. Lastly, consistent with our previous findings[11], oHSV treatment increased CD8+ T cell influx into mice brain tumors which was significantly inhibited when treated with either Fer-1 or Enza (Fig. 7m, n). This suggests that ferroptotic cell death is important to maximize the anti-tumor immune benefit of oHSV therapy.

### Impact of ferroptosis on tumor immune environment after virotherapy

To identify the changes in immune cell landscape dynamics driven by oHSV-induced ferroptosis we treated 005 glioma-bearing mice with oP10 (P) ± Fer-1 to block ferroptosis. CD45+ cells were isolated from tumor-bearing hemispheres 5 days post-oHSV treatment (n = 5 mice/group) and subjected to scRNA-seq using the 10× Genomics platform. We identified a total of 9 distinct clusters of cells comprising distinct populations of microglia, macrophages, monocytes, T cells, neutrophils, natural killer cells, DCs, B cells, and endothelial cells (Figs. 8a and S8a, b). There was not a major change in the composition of the immune cell clusters recruited into the brains of mice treated with oP10 with or without Fer-1. Analysis of cell–cell communication using CellChat analysis inferred cell–cell interaction networks among the clusters and revealed that treatment with Fer-1 changed the interactions between macrophages and T cells

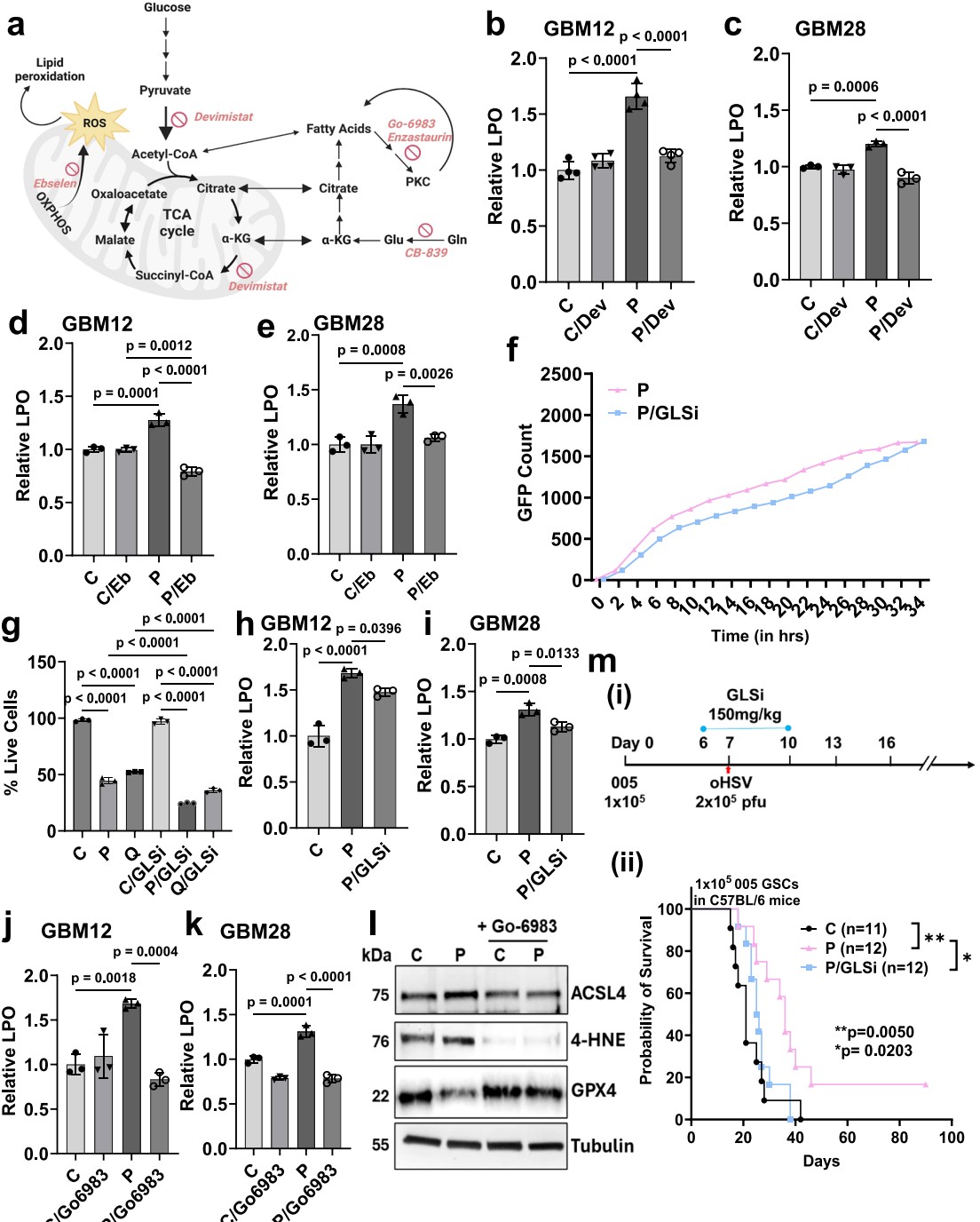

**Fig. 6 | Determinants of oHSV-induced ferroptosis in gliomas. a** Summary of the metabolic pathways altered for glucose and glutamine metabolism by oHSV treatment. Briefly in oHSV-treated cells increased glucose utilization is fluxed into the TCA cycle which increases ROS. Glutamine is shuttled into reductive carboxylation towards fatty acid synthesis. PKC activation in infected cells induced ACSL4 that leads to fatty acid oxidation. Inhibitors used to block individual circuits are labeled in red. [Created in BioRender. Sahu, U. (2025) https://BioRender.com/q78n328]. Changes in relative lipid peroxidation (LPO) in P-infected GBM12 (**b, d**) and GBM28 (**c, e**) cells 48 hpi in the presence of pharmacologic blockade of TCA cycle using Devimistat (Dev) (**b, c**) or by treatment with ROS scavenger, Ebselen (Eb) (**d, e**) compared to untreated control. **f** Kinetics of oHSV replication in GBM12 cells treated with P ± GLSi at an MOI of 0.01. The viral spread of P (pink line) and P + GLSi (blue line) in GBM12 glioma cells was assessed by GFP expression over time using the Cytation 5 live imaging system. GFP object count was quantified and graphed from $n = 3$ wells as the mean. **g** Relative percentage of live cells in oHSVs Q-

or P-infected GBM12 ± GLSi compared to untreated control 48 hpi at MOI 0.01 as accessed by MTT assay. Relative lipid peroxidation in GBM12 (**h**) and GBM28 (**i**) cells treated with P with or without GLSi treatment 48 hpi. Data = mean ± S.D., $n = 3$ independent replicates. Relative levels of lipid peroxidation in GBM12 (**j**) and GBM28 (**k**) cells treated with P ± Go6983 (PKC inhibitor). **l** Western blot for the ASCL4, 4-HNE (end-product of lipid peroxidation) and GPX4 in GBM12 cells infected with P ± Go6983. Tubulin was used as loading control. **m** (i) Experimental design and (ii) Kaplan–Meier analysis of murine 005 GSC tumor-bearing immune-competent mice following treatment with PBS (C) or P or P/GLSi ($n = 11$ for C; $n = 12$ for P; and $n = 12$ for P/GLSi). Log-rank (Mantel–Cox) test. Data = mean ± S.D., $n = 3$ independent replicates. (One-way ANOVA, unpaired two-tailed Student's $t$-test). hpi hours post-infection, C control, P oP10 infected. Source data are provided as a Source Data file.

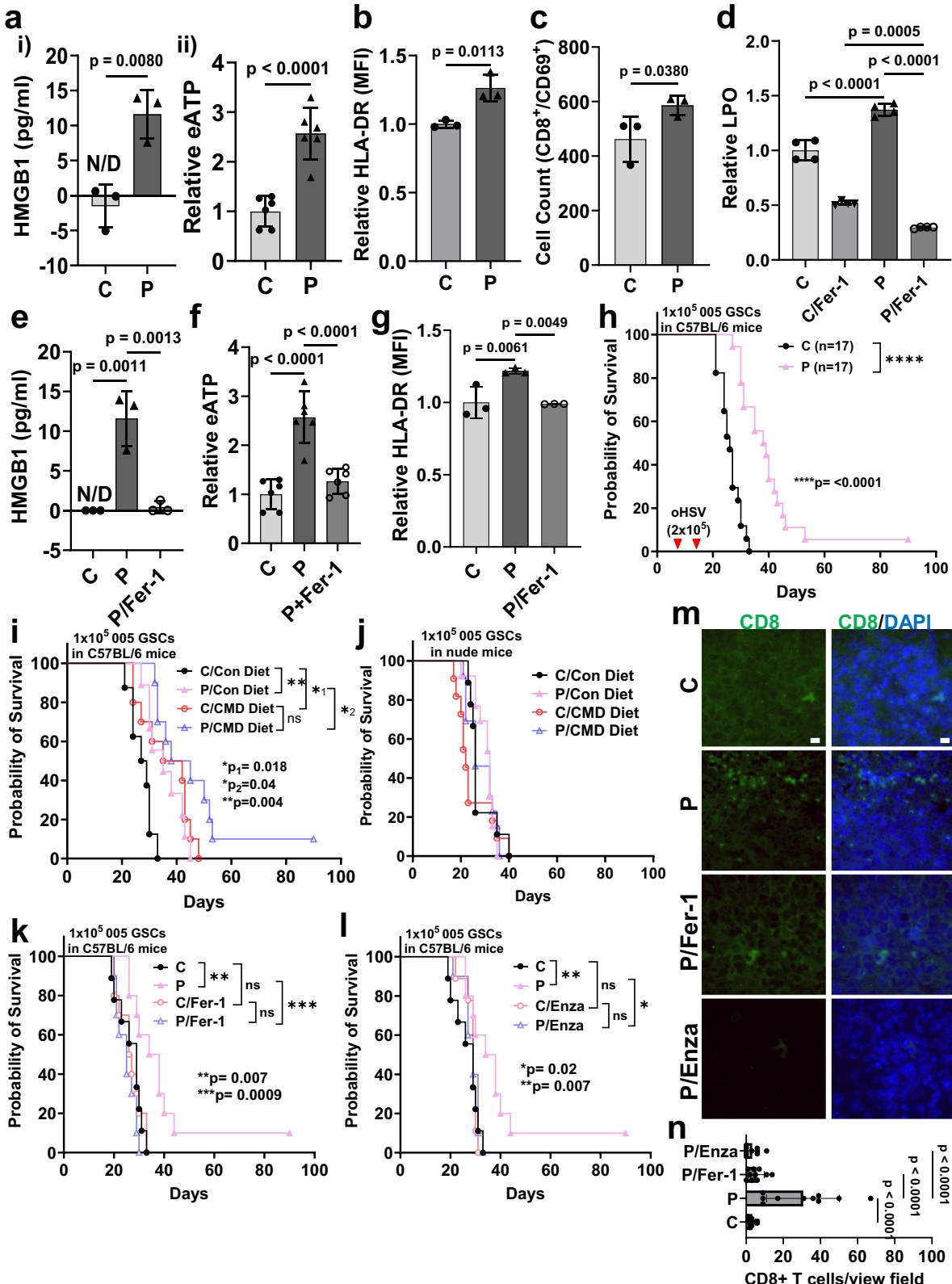

(Fig. 8b, c). Interestingly, immune cells from mice treated with P + Fer-1 had reduced the number of interactions between T cells and other immune cells relative to mice treated with virus alone. There was also a significant change in interaction strengths of the cross talk from macrophages to other immune cells (Fig. 8c). Given the difference in the cell chat we thus examined subclustered T cell populations. There were six distinct subclusters, and cell type identities were assigned to each cluster by canonical markers shown as heatmap: Gzmk+ effector CD8+ T cells, exhausted T cells, gamma delta T cells, naïve T cells, regulatory T cells (Tregs) and natural killer cells (Figs. 8d–f and S8c). Fer-1 treatment resulted in an increase in the Tregs and exhausted T cells with a simultaneous reduction in Granzyme K (Gzmk)+ CD8+ cytotoxic T cells and gamma delta T cells (Fig. 8e). This showed that ferrostatin-treated mice had a shift

**Fig. 7 | oHSV-induced ferroptosis is important for anti-tumor immune therapy.**
**a** Levels of HMGB1 (i) and eATP (ii) in conditioned medium of P-infected GBM12 cells relative to control (mean ± S.D.). **b** PBMC-derived DCs were incubated with supernatants from control or infected GBM cells for 48 h. CD11c⁺ DC activation was analyzed by cell surface human leukocyte antigen (HLA)-DR expression by flow. Data = relative MFI in infected cells ± S.D. **c** Antigen presentation ability of the DCs assessed by increased proliferation of CD8⁺/CD69⁺ T cells after 5-day culture with educated DCs from B (mean ± S.D.). Lipid peroxidation in cell lysates (**d**), HMGB1 released (**e**) and relative eATP released (**f**) in infected cell supernatant ± Fer-1.
**g** HLA-DR expression on CD11c⁺ DCs treated with infected cell supernatant ± Fer-1 by flow. **h** Kaplan−Meier analysis of murine 005 glioma-bearing mice following treatment with PBS (C) or P in C57BL/6 immunocompetent mice ($n = 17$/group). 7- and 14-day post $1 \times 10^5$ 005 GSCs implantation, mice were injected with either C or $2 \times 10^5$ P and monitored for survival. (Log-rank (Mantel−Cox) test, ****$P < 0.0001$). Kaplan−Meier survival curves comparing C- and P-treated C57BL/6 (**i**) or nude (**j**)

mice orthotopically injected with $1 \times 10^5$ 005 glioma under control (Con) or cysteine-depleted methionine restricted (CMD) diet ($n = 8$, C/Con; $n = 9$, P/Con; $n = 10$, C/CMD and P/CMD). Kaplan−Meier survival analysis of murine 005 glioma-bearing C57BL/6 mice following ± oP10 and Fer-1 (10 mg/kg; i.p.) (**k**) or Enzasturin (Enza; 30 mg/kg; oral gavage) (**l**) treatment blocking ferroptosis and PKC activation, respectively. Values from a single experiment with C and P groups, the same in (**k**, **l**). $n = 9$ for C and C/Enza and $n = 10$ for P, P/Enza, C/Fer-1 and P/Fer-1. Log-rank (Mantel−Cox) test, *$P < 0.05$, **$P < 0.01$. Representative 100× images of CD8 from sagittal sections of mouse brains 5-day post-virus injection ($n = 3$ mice/group; images from at least 2 different fields/section); Scale = 10 μm (**m**) and quantification of tumor-infiltrated CD8⁺ T cells (**n**). Data = mean ± S.D., $n = 3$ independent replicates for A-G. (One-way ANOVA, unpaired two-tailed Student's *t*-test). N/D not detectable, C control, P oP10 infected. Source data are provided as a Source Data file.

towards immune suppressive T regs from active Gzmk +ve T cell phenotypes.

To evaluate the importance of ferroptosis in T cell functionality we evaluated the number of tumor and virus antigen-specific CD8 T cells as described[1]. Briefly, mice bearing 005-OVA glioma were treated with oP10 (P) ± Fer-1. Fifteen days post-virus treatment mice were sacrificed and tumor-bearing hemispheres were analyzed for OVA and gB recognizing T cells by tetramer analysis. Flow cytometry revealed a significant increase in OVA⁺/CD8⁺ T cell count with oP10 treatment which was inhibited by blocking ferroptosis with Fer-1 treatment (Figs. 8g and S8d, e). Interestingly, there was no significant difference in antivirus gB recognizing T cells (Figs. 8h and S8f). Collectively this shows that ferroptosis is important for activation of anti-tumor T cells after oHSV treatment.

### Effect of mutant IDH on virus efficacy

Collectively, analysis of the transcriptomic data from patient tumor specimens after virotherapy and our in vitro and in vivo analysis shows that oHSV treatment drives glucose utilization into the TCA cycle and glutamine metabolism towards reductive carboxylation to increase fatty acid synthesis. Changes in cellular metabolism have an impact on cancer growth and in particular mutations in IDH enzyme dramatically affect glioma biology and prognosis, and cells harboring mutant IDH cannot undergo reductive carboxylation[26,27]. Interestingly in the NCT03152318 trial, patients with wildtype IDH (wtIDH) had a better response to oHSV than patients harboring mutant IDH[8]. Thus, to evaluate the significance of oHSV treatment on mutant IDH glioma we compared matched murine glioma cells harboring heterozygous mutant IDHR132H (IDHR132H) to wtIDH glioma[27]. Figure 9a, b shows that virus treatment induced a significant increase in oxygen consumption rate (OCR) indicative of enhanced mitochondrial respiration in wtIDH glioma cells, but not in IDHR132H glioma. Increased levels of lipid peroxidation and ROS (cellular and mitochondrial) were observed in wtIDH glioma cells but were either unaltered or reduced in IDHR132H glioma cells (Figs. 9c−h and S10). The IDHR132H neomorphic mutation confers a gain-of-function activity resulting in the reduction of α-KG to the oncometabolite 2-hydroxyglutarate (2-HG) (Fig. 9i)[28,29]. Thus, this frequently occurs as a heterozygous mutation in glioma patients. The cell lines used here also retain this heterozygosity and thus when IDHR132H glioma cells are treated with AGI-5198 (a selective IDHR132H inhibitor[27,30–32]), wtIDH allele can restore the balance of α-KG to permit reductive carboxylation eventually leading to enhanced lipid peroxidation[33]. Treatment of IDHR132H glioma cells with AGI-5198 resulted in increased lipid peroxidation following oHSV treatment (Fig. 9j).

This implied that inhibition of IDHR132H in conjunction with viro-immune therapy would induce ferroptosis and improve immune efficacy. To evaluate if AGI-5198 could also have a therapeutic benefit in conjunction with virotherapy we evaluated response to oP10 in

immune-competent mice bearing IDHR132H tumors. While virotherapy did not have an impact on the survival of mice bearing IDHR132H glioma (Fig. S9a, b), treatment of these mice with AGI-5198 significantly improved the therapeutic benefit (Figs. 9k and S9c, d). In a second humanized immune-competent mouse model[34] bearing U87-mIDH glioma, treatment with oP10 showed significantly reduced tumor growth when treated in combination with AGI-5198 as compared to virotherapy alone (Figs. 9l, m and S9e). Collectively, our data show that IDHR132H high-grade glioma patients treated with oHSV will likely benefit from a combination with inhibitors that block IDHR132H activity.

## Discussion

Brain tumors, like most other cancers, rewire cellular metabolic circuits to maintain energy production and redox homeostasis, as well as for the generation of building blocks to support cell division[35]. These metabolic adaptations have a dramatic effect on tumor evolution, and a better understanding of these changes will provide unique opportunities for therapeutic intervention. The identification of gain of function mutations in IDH and their link with prognosis in GBM patients was a landmark discovery that has altered the ways in which GBM is diagnosed and managed today[36]. Brain tumors are addicted to glutamine that is converted to αKG to serve as fuel for the TCA cycle in wtIDH cells. Mutant IDH converts αKG to 2-HG, an oncometabolite that cannot be used as a TCA metabolite and hence significantly alters the metabolic profile of tumors[37]. While high-grade glioma patients bearing mutant IDH have a favorable prognosis, the impact of IDH status in response to virotherapy is not known.

Here we investigated the impact of oHSV immunotherapy on glutamine flux. Our results show that oHSV treatment of wtIDH glioma diverts glutamine from entering the TCA cycle towards reductive carboxylation and lipid biosynthesis. This generates more polyunsaturated fatty acids (PUFA), whose oxidation culminates in lipid peroxidation[38]. The degree of accumulation of lipid peroxidation products resulting from oxidative degradation of lipids regulates ferroptosis, an iron-dependent immunogenic form of regulated cell death[39–41]. Our results show that the induction of ferroptosis in wtIDH glioma is key to reaping immune benefits from virotherapy, and ferroptosis inhibitors abrogate virus-induced therapeutic efficacy in syngeneic glioma-bearing mice. HSV-1 infection has been shown to induce ferroptosis in vitro and in vivo in cultured human astrocytes, microglia, and murine brains[42], however, the importance of ferroptosis in determining the efficacy of onco-virotherapies like oHSV has not been evaluated.

Since tumors bearing mutIDH are incapable of reductive carboxylation[26], we evaluated the impact of IDHR132H on virotherapy. Our results showed that glioma cells harboring mutIDH do not undergo ferroptosis and inhibition of IDHR132H function synergized with virus to improve therapeutic benefit in immune-competent mice.

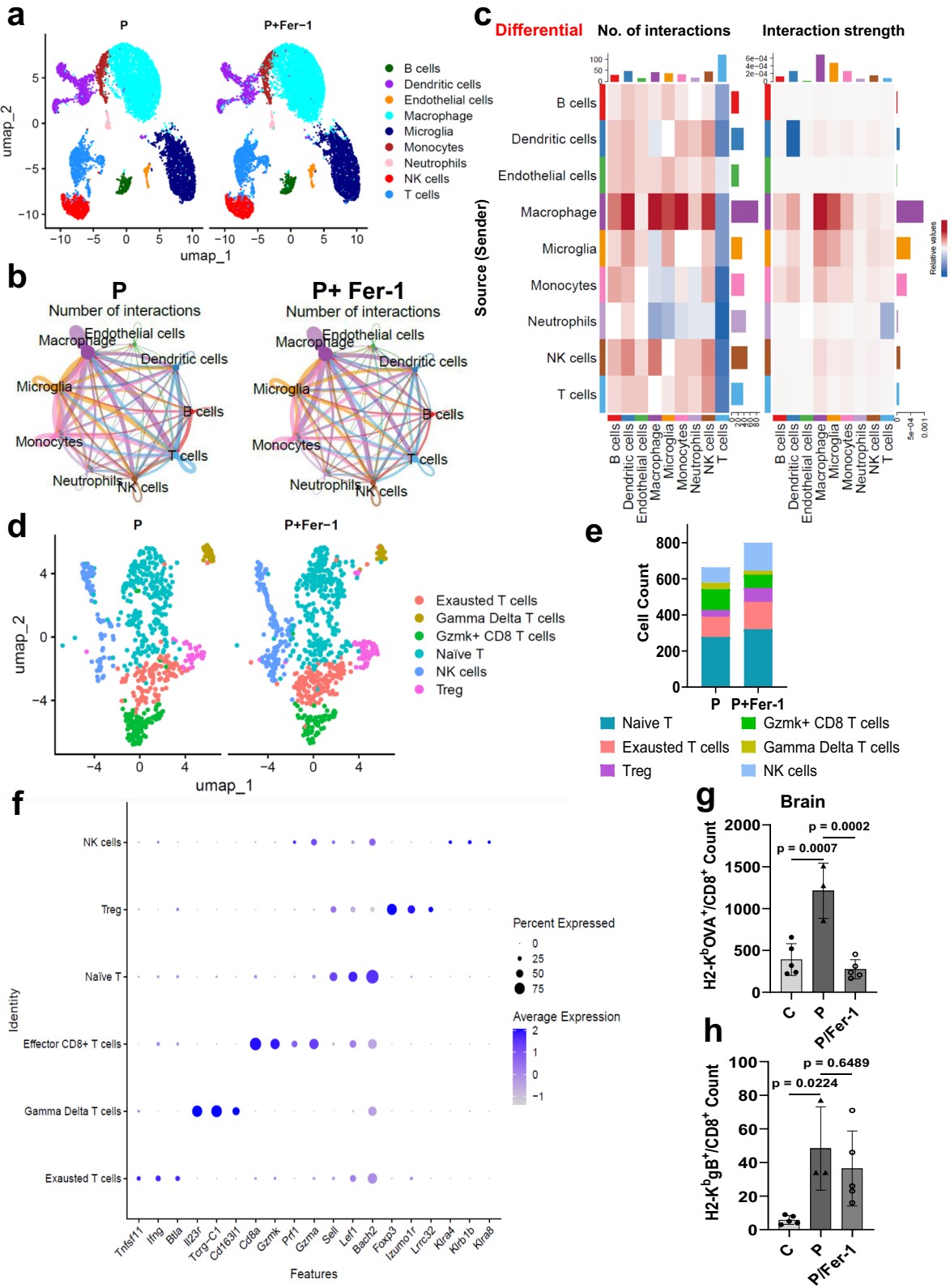

Consistent with this, the inhibition of mutIDH has been shown to sensitize glioma-bearing mice to PD-L1 blockade[27]. Together these results indicate that mutIDH gliomas are resistant to immunotherapy and that this can be overcome by pharmacologic inhibition of mutIDH function. It is interesting to note that contrary to these findings, mutIDH has also been described to support erastin-induced ferroptosis in vitro, implying that the impact of IDH status on ferroptosis

could be context-dependent[43]. Consistent with our observations, the activation of immunity has been linked to the survival of glioblastoma patients treated with CAN-3110 (an oncolytic HSV therapy)[8]. It is interesting to note that in this study the median survival times of IDH wt rGBM patients was 10.9 months and only 5.4 months for patients harboring the mutant enzyme[8]. While the small number of patients makes it hard to evaluate the significance, it is interesting to speculate

**Fig. 8 | Impact of Ferroptosis on tumor immune environment after virotherapy.** Single-cell sequencing analysis of CD45[+] cells isolated from 005 glioma-bearing hemispheres of mice treated with oP10 5 days after virotherapy. **a** UMAP visualization of the composition of the annotated cells from virus or virus and ferrostatin-1 treated animals, each color representing a different cell cluster. Cell number: 10504 for P; 12427 for P+Fer-1. **b** Cell−cell interaction network among the major cell clusters in P and P+Fer-1 group. Line width represents number of interactions between the two interacting cell types. **c** Heatmap comparing differential number of interactions and interaction strength between the immune cell populations in tumors of P and P+Fer-1 treated mice. P+Fer-1 has fewer interactions of all cell types to T cell cluster. **d** UMAP represents subclusters of T cells, each color

denoting an individual cluster. Cell number: 664 for P; 801 for P+Fer-1. **e** A composite bar graph of the T cell cluster representing altered T cell subclusters in P and P+Fer-1. **f** Marker genes used to annotate T cell subclusters. Dot size depicts the % of cells of a cluster expressing the given gene, and color intensity indicates the expression level of the gene by that cluster. **g, h** Mouse glioma 005-OVA tumors were established in C57BL/6 mice. Seven days later, tumor-bearing mice were treated with P±Fer-1. Anti-tumor and antiviral-specific T cells were analyzed by OVA tetramer (**g**) and HSV gB tetramer (**h**) staining in tumors 21 days post-tumor implantation (C and P/Fer-1: $n = 5$; P: $n = 3$), respectively. Data = Mean cell count ± S.D from $n = 3–5$ mice per group. (One-way ANOVA). C control, P oP10 infected. Source data are provided as a Source Data file.

that the combination of virotherapy with AGI-5198 should synergize in patients harboring mutIDH. In fact, our in vivo studies support the use of drugs that can block mutIDH function in conjunction with oHSV therapy to improve benefit.

Apart from glutamine, most solid tumors including glioma also hijack glucose metabolism to promote aerobic glycolysis (Warburg effect) even in the presence of oxygen[44]. Using U[13]C labeled glucose flux analysis we observed an increased flux of glucose-derived metabolites into the TCA cycle, bypassing the Warburg effect. Consistent with our findings, HSV-1 encoded viral protein UL16 has been shown to promote oxidative phosphorylation of glucose along with increased ATP production[45]. Glucose utilization has also been reported to be important for viral infection and replication in vitro[46,47]. On the other hand, infection of Vero cells with wt HSV-1 has been shown to increase aerobic glycolysis promoting the Warburg effect[47]. It is important to note that while this study reported increased ATP synthesis, the impact of viral infection on glucose flux was not studied. While inhibition of glycolytic pathways in vitro attenuates virus replication, its inhibition in vivo also deprives immune cell function and impairs T cell activity resulting in weak virus clearance and encephalitis[48,49]. Our results show that increased mitochondrial activity increased cellular ROS which contributed to oxidative stress. The finding that inhibition of pyruvate to acetyl CoA with Devimistat blocked glucose flux into the TCA cycle and inhibited lipid peroxidation (hallmark of ferroptosis) underscores the importance of rewiring of both glucose and glutamine metabolism in infected cells for induction of ferroptosis. Using a pan-kinome functionality assay we also observed activation of PKC, a class of serine/threonine kinases that acts as a pleiotropic regulator of cell proliferation, differentiation, and survival[50]. PKC activation has also been shown to be an important sensor of lipid peroxidation to induce ferroptosis in a breast cancer model[17], and a close link between PKCs and oxidative stress is well recognized[51–54]. Here, we observed that PKC activation was important to induce executors of lipid peroxidation.

Collectively, our results indicate that oHSV orchestrates increased glucose utilization into the TCA cycle leading to the induction of oxidative phosphorylation and ROS. At the same time, infected glioma cells also shuttle glutamine into reductive carboxylation, increasing lipid synthesis that generates more DAG leading to PKC activation. Together, this induces lipid peroxidation and ferroptosis. Analysis of transcriptomic changes in patient bulk RNA sequencing pre- and post-G207 also uncovered a significant enrichment of pathways related to oxidative phosphorylation and ferroptosis (Fig. S1). Interestingly, in rGBM patients treated with CAN-3110, we also observed a significant enrichment of oxidative phosphorylation and mitochondrial respiratory chain complex activity in HSV seropositive patients but not in HSV seronegative patients. It is interesting to note that some patients did not convert to become HSV-1 seropositive even after virus treatment, which correlated with significantly worse response to treatment[8]. While the reason why these patients did not convert is unclear, it can be speculated that this might reflect a highly immune-suppressed state of these patients. This implies that changes in tumor metabolism after virotherapy might provide a unique opportunity to predict patients

likely to benefit from virotherapy and mount a subsequent immune response. In the future, a more detailed exploration of the mechanisms by which oHSV manipulates cellular metabolic pathways might provide targets to improve therapeutic outcomes. Our results support the use of IDHR132H inhibitors in conjunction with virotherapy for high-grade mutIDH glioma patients being treated with oHSV.

## Methods

### Cell line isolation and culture

Patient-derived primary GBM cells (GBM12 and GBM28) were kindly provided by Dr. Jann N. Sarkaria (Mayo Clinic, Rochester, USA)[55]. GBM cells were cultured in either neurosphere culture media (neurobasal medium supplemented with 2% B27 minus vitamin A, 20 ng/mL human epidermal growth factor (EGF) and 20 ng/mL basic fibroblast growth factor (FGF)) or in adherent conditions (DMEM supplemented with 2% FBS, penicillin, streptomycin, and plasmocin; 2F DMEM). Monkey kidney epithelial-derived Vero cells were purchased from ATCC and cultured in DMEM with 10% FBS, penicillin, streptomycin, and plasmocin. U87-mIDH1[R132H] (U87-mIDH) cells were purchased from ATCC (HTB-14IG-LUC2) and maintained in DMEM supplemented with 10% FBS, penicillin, streptomycin, plasmocin, and 0.8 μg/mL blasticidin. Murine 005 cells were obtained from Dr. Inder Verma (Salk Institute for Biological Studies, CA, USA). Mouse IDH1 wildtype (wtIDH) and IDH1[R132H] mutant (IDHR132H) cells were a kind gift from Dr. Maria Castro (University of Michigan, MI, USA)[56]. Mouse wtIDH, IDHR132H, and 005 GSCs were cultured as spheres in DMEM/F12 media, supplemented with 2 mM L-glutamine, 1% N2 supplement (for wtIDH and IDHR132H cells), 0.5% penicillin-streptomycin, recombinant human epidermal growth factor (EGF) (20 ng/mL) and recombinant human FGF-basic (20 ng/mL) and dissociated with accutase for passaging. GBM12 and GBM28 cells were authenticated by the University of Arizona Genetics Core via short tandem repeat (STR) profiling and maintained below passage 40 afterward. All cells are negative for Mycoplasma and checked routinely (Mycoplasma PCR Detection Kit, #G238, Applied Biological Materials, Canada). We used oHSV-Q (Q) and oHSV-P10 (oP10 or P) for this study. Both the viruses were propagated in Vero cells, and the plaque-forming units per milliliter used in this study were determined by virus titration in Vero cells as previously described[10]. Cells were infected with Q or P at an MOI 0.01 (GBM12, and 005) and MOI 0.03 (GBM28, wtIDH, IDHR132H, and U87-mIDH) for 1 h and then media were replaced with 2F DMEM. Cells were collected 48 hours post-infection (hpi) for different assays unless otherwise mentioned.

Glioma cells were treated with chemicals including the ferroptosis inhibitors Ferrostatin-1 (Fer-1; 10 μM, Cat. No. HY-100579, MedChemExpress, USA), PKC inhibitors Go6983 (5 μM, Cat. No. S2911, Selleckchem, USA) and enzastaurin (5 μM, Cat. No. S1055 Selleckchem, USA); TCA cycle inhibitor Devimistat (10 μM MedChemExpress, USA); ROS scavenger Ebselen (25 μM, Cat. No. HY-13750 MedChemExpress, USA), IDHR132H inhibitor AGI-5198 (1.5 μM, Cat. No. HY-18082, MedChemExpress, USA) and glutaminase inhibitor CB-839 (2uM, Cat. No. HY-12248, MedChemExpress, USA). All the inhibitors were

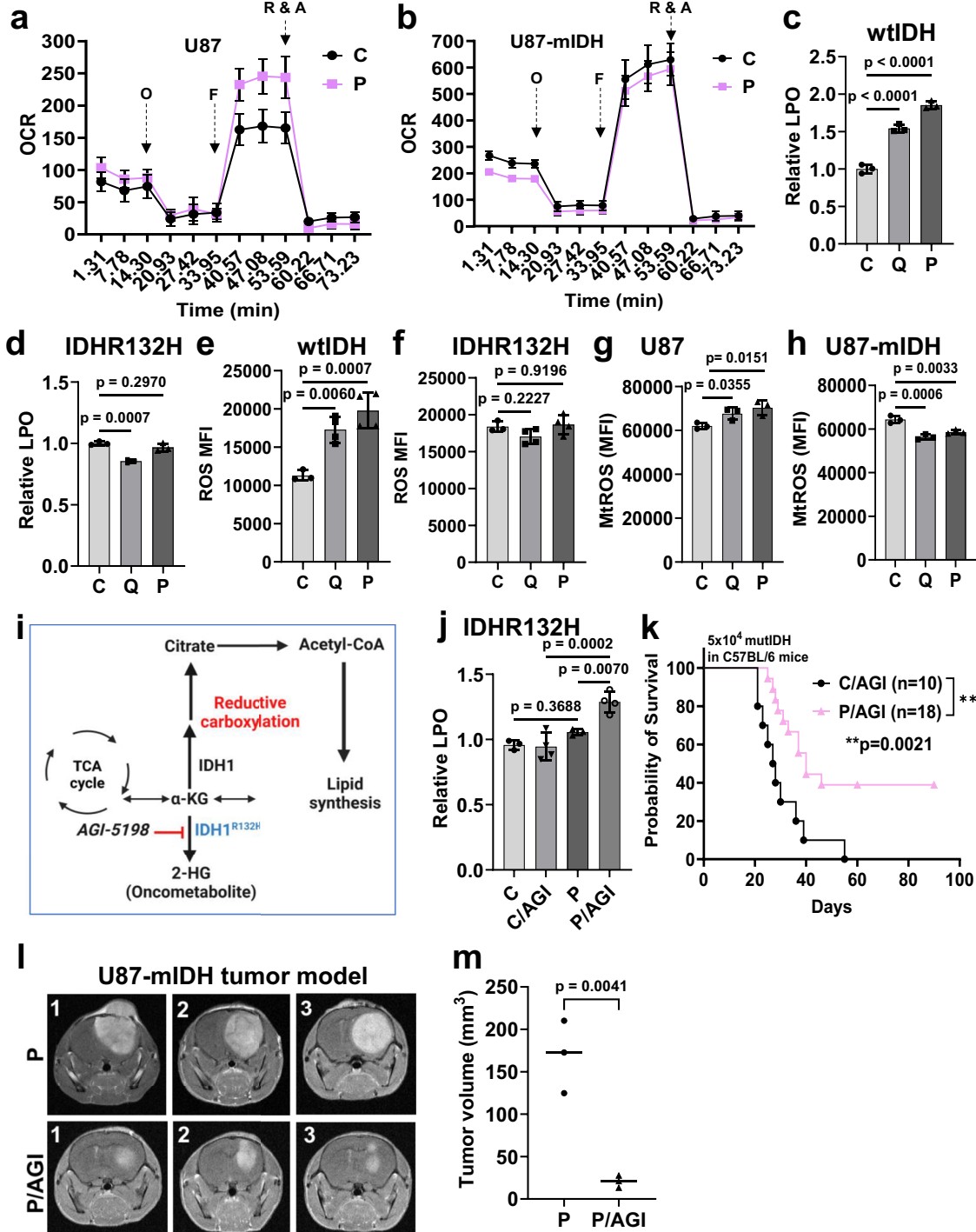

**Fig. 9 | Inhibition of reductive carboxylation abrogates oHSV-induced immunity. a, b** Seahorse Bioscience XFe96 extracellular flux analyzer was used to measure oxygen consumption rate (OCR) in pMoles/min in uninfected and oHSV-infected U87 and U87-mIDH cells 24 hpi (MOI = 0.03). OCR of C or P treated cells in pmol/min/8 × 10⁴ in U87 cells (**a**) and U87-mIDH cells (**b**) in real-time under basal conditions and in response to mitochondrial inhibitors (oligomycin (O); FCCP (F); rotenone (R) and antimycin (A)), n = 6/group. Relative lipid peroxidation levels in oHSV-infected murine wtIDH cells (**c**) and IDHR132H cells (**d**) following oHSV infection compared to uninfected controls (n = 3). Intracellular ROS (**e, f**) (C: n = 3; Q/P: n = 4) and mitochondrial ROS (**g, h**) (n = 3) in oHSV-infected wtIDH and IDHR132H glioma cells. **i** Schematic to block reductive carboxylation via IDHR132H.

[Created in BioRender. Mullarkey, M. (2025) https://BioRender.com/n64s900]. **j** Relative levels of LPO in IDHR132H cells upon oHSV infection ± inhibitor of IDHR132H (AGI-5198) (n = 3 for C and P; n = 4 for C/AGI and P/AGI). **k** Kaplan–Meier analysis of murine IDHR132H tumor-bearing C57BL/6 mice following treatment with AGI-5198 (AGI) with or without P treatment (n = 10 for C/AGI group and n = 18 for P/AGI group). Log-rank (Mantel–Cox) test. Brain magnetic resonance imaging (MRI) (**l**) and quantification of tumor volume (**m**) of U87-mIDH tumor-bearing NSG mice treated with P ± AGI-5198. (Unpaired two-tailed Student's t-test; n = 3/group). (One-way ANOVA, t-tests). hpi hours post-infection, C control, P oP10 infected. Source data are provided as a Source Data file.

reconstituted to at least 1000× concentration following the manufacturer's protocol.

## Animals

All procedures were reviewed and approved by the Augusta University Institutional Animal Care and Use Committee (IACUC, Protocol # 2022-1080). Male and female mice were used for all experiments. Wildtype C57BL/6 mice (strain #000664), (Foxn1$^{nu}$/Foxn1$^{nu}$) athymic nude mice (strain #002019) and (NOD.Cg-PrkdcscidIL2rgtm1Wjl/SzJ) NSG mice, (Strain #005557) were obtained from Jackson Laboratories. Mice were housed under standard guidelines in pathogen-free conditions with *ad libitum* access to food and water at controlled temperature (21 ± 2 °C), humidity (55-60%), and 12 h light/dark cycle. The research conducted in this study complies with all relevant ethical regulations for animal testing and research.

## Viral and orthotopic cell injections

For orthotopic cell transplantation experiments, 6-week-old C57BL/6 and nude male and female mice were injected with 100,000 murine 005 cells, 6-week-old C57BL/6 mice were injected with 50,000 murine mutIDH cells and 6-week old NSG mice were injected with 100,000 human U87-mIDH cells each resuspended in 2 µL PBS at the following coordinates (with bregma as reference): 2 mm lateral, 1 mm rostral and 3.5 mm deep, and at a flow rate of 0.4 µL/min. Day 7 post-tumor implantation mice were randomized to be intratumorally injected with either 2 µL PBS or 2 × 10$^5$ pfu oHSV in a final volume of 2 µL. Mice were monitored daily for signs of tumor morbidity and sacrificed when their body score reached 2 or below.

## Diet allocation

Special diets were created by LabTest Diet (Lab Supply, Texas, US). A control (Con) diet with 0.43% methionine and 0.33% cystine (w/w, Item code. 1816296-203, LabSupply) and a cystine-depleted methionine restricted (CMD) diet with 0.15% methionine and 0.0% cystine (w/w, Item code. 1818896-203, LabSupply) were used[25]. C57BL/6 and nude mice were subjected to the diet seven days post murine 005 GSCs tumor implantation. Investigators were aware of diet allocation or outcome assessments during experiments. Day 3 and day 7 post-diet transition mice were intratumorally injected with either 2 µL PBS or 2 × 10$^5$ pfu oHSV in a final volume of 2 µL. Mice were fed with the special diet till the termination of the experiment.

## Drug/Inhibitor dosage

**Ferrostatin-1 and enzastaurin**. Day 7 and day 14 post 005 GSC tumor implantation, C57/BL6 mice were intratumorally injected with either 2 µL PBS or 2 × 10$^5$ pfu oHSV in a final volume of 2 µL. Mice were treated with Ferrostatin-1 (20 mg/kg; intraperitoneally) or Enzastaurin (30 mg/kg; oral gavage) for four consecutive days starting from the day of virus injection.

**AGI-5198**. Seven days after murine IDHR132H tumor implantation, C57BL/6 mice were split into two cohorts each of intratumoral injection with either PBS or 2 × 10$^5$ pfu oHSV in a final volume of 2 µL followed by intraperitoneal injection with either vehicle (10% DMSO + 90% Corn oil) or 40 mg/kg AGI-5198 (MedChemExpress, USA) thrice every alternate day for 3 cycles with a 2-day interval in between[27].

**CB-839**. Day 6 post 005 GSC tumor implantation, mice were split into two cohorts each of vehicle (10% DMSO + 90% Corn oil) or 150 mg/kg glutaminase inhibitor (CB-839, GLSi) orally for 5 consecutive days. Each cohort was split into 2 subgroups each of intratumoral injection with either PBS or 2 × 10$^5$ pfu oHSV in a final volume of 2 µL.

**U87-mIDH xenograft model injected with PBMC**. To mimic the human immune system in NSG mice (female), 4 and 11 days after U87-

mIDH tumor implantation, we intraperitoneally (i.p.) injected each mouse with peripheral blood mononuclear cells (PBMCs) isolated from a single healthy donor using a buffy coat (Shepeard Community Blood Center, Augusta, Georgia, USA) by Ficoll gradient centrifugation. On days 6 and 13 of tumor implantation, we i.p. delivered activated T cells derived from the same donor to each mouse. Dynabeads human T-Activator CD3/CD28 for T Cell Expansion and Activation (Cat. No. 11132D, Thermo Fisher Scientific, USA) were used to activate human T cells in RPMI-1640 media supplemented with 20% FBS, 100 U/mL Penicillin/Streptomycin and 30 ng/mL human IL-2 (Cat. No. 200-02, PeproTech, USA) for 3 days. Days 7 and 14 after tumor implantation, each mouse received i.c. oP10 or saline ± AGI-5198 (regime as discussed earlier). Magnetic resonance imaging (MRI) was done on day 35 post-tumor implantation.

## mRNA sequencing analysis

**G207 study[7]**. From log fold change obtained from differential expression analysis, GSEA was performed against GO and KEGG pathways. The clusterProfiler package was used for the analysis and plotting with R version 4.2.2.

**dbGap study[8]**. To see the shift in gene profile between the pre- and post-oHSV groups, the differential expression analysis was performed using the limma package. From log fold change obtained from differential expression analysis, GSEA was performed against GO and KEGG pathways. The clusterProfiler package was used with R version 4.2.2.

## Single-cell sequencing (sc-seq)

To perform the scRNA-seq experiment, CD45+ immune cells were isolated from 005 GSC tumor-bearing hemisphere of female C57BL/6 mice 5 days post-oHSV treatment (10X Genomics Inc. Pleasanton, CA). Cells were resuspended in PBS (without Ca or Mg) with 0.04% BSA and freshly processed. Cell viability was assessed with Cellometer Auto 2000 (Nexcelom) using an AOPI staining. Samples with viability ≥80% were loaded in the Chromium X instrument (10X Genomics) to capture ~10 × 10$^3$ targeted cells using Chromium Next GEM Single Cell 3′ Reagent Kit v3.1 Dual Index form 10X Genomics (cat# 1000268). scRNA-seq libraries were generated according to the manufacturer's instructions. QC (quality control) analysis was performed prior to sequencing with 2100 Bioanalyzer and 2200 TapeStation (Agilent, Santa Clara, CA, USA). Library concentration was assessed prior to sequencing using a Qubit Fluorometer (Thermo Fisher). Sequence was performed with Novaseq6000 System Illumina (Illumina Inc. San Diego, CA) platform following 10X Genomics guideline; read lengths U28|I10|I10|Y90 and sequencing depth 40,000 paired-end reads per cell. Samples were processed at the Integrated Genomics Core at Augusta University Georgia Cancer Center.

The raw sequencing reads were processed and aligned to the GRCm39 mouse genome using CellRanger (v7.0.0). Downstream analyses were conducted in R (v4.4.1) using Seurat (v5.1.0). Quality control filtering retained cells with more than 200 detected genes, less than 20% mitochondrial gene expression, and excluded cells with abnormally high unique molecular identifier counts to remove low-quality cells. Additionally, the DoubletFinder package (v2.0.4) in R was used to remove doublets. The data were normalized with SCTransform, and highly variable genes were identified to enhance downstream analysis. To mitigate batch effects across samples, Seurat objects were integrated using canonical correlation analysis. Cells were clustered using the Louvain algorithm, with a clustering resolution set to 0.5 to define distinct cell populations. Cell types were then annotated with the scType tool, utilizing a custom marker gene list optimized for mouse brain and cancer cell types. Subsequently, the CellChat package (v2.1.2) in R was utilized to infer intercellular communication patterns among these annotated cell types. This workflow

provided a comprehensive approach to identify and interpret cell-type-specific interactions both within and between the datasets.

## MitoTracker™ Red staining

Cells were seeded in 6-well plates at a density of $5 \times 10^5$ cells/well. After an 18-h incubation, cells were labeled with 25 nM MitoTracker Red CMXRos (Cat no. M7512, Thermo Fisher Scientific, USA) dissolved in DMEM for 45 min at 37 °C. Cells were washed once with PBS. Bright field and fluorescent images were captured using a Nikon Ts2 microscope (Nikon, Tokyo, Japan).

## mtDNA copy number

$5 \times 10^5$ GBM12 cells were seeded per well in a 6-well plate. Cells were infected with oHSVs Q and P (MOI 0.01), media was replaced 1 h post-infection and cells were collected 24 hpi. Genomic DNA was isolated using the Qiagen DNeasy Blood and Tissue Kit. Briefly, a SYBR-based qPCR with primers specific to the tRNA region of mtDNA and β2-microglobulin region of nuclear DNA was performed to calculate the mtDNA copy number using a previously published method[57].

## Electron transport chain (ETC) analysis

For the ETC activity assay, activities of ETC complexes were measured by kinetic spectrophotometric assays as previously described[58,59]. $5 \times 10^6$ cells were harvested by trypsinization. After sonication of cell pellets, protein concentration was determined by Bradford assay. The kinetics of each OXPHOS complex and citrate synthase (CS) from the TCA cycle were measured using a Tecan Infinite M200 microplate plate reader. The assay is based on the measurement of oxidation/reduction of substrates or substrate analogs of individual complexes. The enzyme activity was normalized to the protein concentration.

## ROS measurement using CellROX deep red reagent staining and flow cytometry

GBM cells were seeded in 12-well plates at a density of $2 \times 10^5$ cells/well, infected with oHSV at different MOIs as mentioned earlier. 48 h after oHSV infection, cells were labeled with 5 µM CellROX deep red reagent (Cat. No. C10422, ThermoFisher Scientific, USA) dissolved in DMEM media for 30 min. Cells were washed and prepared for flow cytometry. Flow cytometry was performed using a Novocyte Quanteon flow cytometer using an APC filter to detect ROS signals. Data was processed using FlowJo v10 software. Cells were gated first by forward and side scatter areas (FSC-A and SSC-A) to remove the debris. The intensity of H2DCFDA of cells in the gate was visualized in the APC channel and represented as MFI and replicates for each condition were plotted using GraphPad Prism 10 software.

## Mitochondria ROS measurement by MitoSOX red staining and flow cytometry

We utilized mitochondria-targeted MitoSOX dye to measure mitochondrial superoxide. $2 \times 10^5$ glioma cells were seeded per well in 12-well plates, infected with oHSV at different MOIs (MOI 0.01 for GBM12 and MOI 0.03 for U87 and U87-mIDH cells). 48 h after oHSV infection, cells were labeled with 5 µM MitoSOX deep red reagent (Cat. No. M36008, ThermoFisher Scientific, USA) dissolved in DMEM media for 30 min. Cells were washed and prepared for flow cytometry. Novocyte Quanteon flow cytometer was used to detect ROS signals in the APC filter and data was processed using FlowJo v10 software. To remove the debris cells were gated first by FSC-A and SSC-A. MitoSOX positive cells in the gate were represented as MFI and replicates for each condition were plotted using GraphPad Prism 10 software.

## Glutathione measurement

GSH concentration was calculated using the Glutathione Colorimetric Detection Kit (Cat. No. EIAGSHC, ThermoFisher Scientific, USA) as detailed in the manufacturer's protocol. Briefly, $1 \times 10^6$ glioma cells were plated per well of a 6-well plate and grown overnight. The next day, cells were infected with oHSV (MOI 0.01 for GBM12 and MOI 0.03 for GBM28 and wtIDH and mutIDH cells) for an hour followed by treatment with different inhibitors. 48 h post-treatment, cells were washed with cold PBS and lysed with 100 µL ice-cold 5% 5-sulfo-salicylic acid dehydrate. Cleared supernatant was used to measure glutathione (GSH) and oxidized glutathione (GSSG). Samples and standards were treated with 2-vinylpyridine for 1 h for GSSG measurement. Values were interpolated from the standard curve and reduced GSH concentrations were determined by subtracting the GSSG concentration from values obtained from non-treated samples and standards and represented relative to uninfected control samples.

## Stable isotope-labeled metabolite analysis

For glucose and glutamine flux measurement, $2 \times 10^6$ cells were seeded per 10 cm dish ($n = 6$/group). 14 h after infection of GBM12 cells with oHSVs (MOI 0.01), media was replaced with glucose or glutamine-free media for 4 h. Cells were supplemented with 2F DMEM containing U-13C glucose (Cat. No. CLM-1396-PK, Cambridge isotope laboratories, USA) or U-13C glutamine (Cat. No. CLM-1822-H-PK, Cambridge isotope laboratories, USA) for 6 h as described earlier[60]. Cells were collected by trypsinization, washed with PBS and $1.3 \times 10^6$ cells/sample were pelleted and frozen at −80 °C. The cells were freeze-thawed in liquid nitrogen, and then sonicated in methanol: water (50:50 v/v). The cell metabolites were extracted using the liquid-liquid extraction method and analyzed through Liquid Chromatography-Mass Spectrometry as described previously[60–64]. The TCA and glycolysis metabolites and their intermediates were separated using Luna 3 µM NH2 (100 Å) HPLC column, whereas mobile phase was A and B in 5 mM ammonium acetate in water (pH 9.9) and acetonitrile, respectively. Mass spectrometry data were acquired via multiple reaction monitoring using a 6495 Triple Quadrupole mass spectrometry coupled to an HPLC system (Agilent Technologies, Santa Clara, CA) Agilent Mass Hunter Software. The detailed conditions and parameters of the LC-MS are described previously[65]. The acquired data were analyzed and integrated into each peak using Agilent Mass Hunter Quantitative Analysis software. The percentage of metabolites incorporation (13C) was calculated using Microsoft Excel from peak area and represented as either bar graphs or as a heatmap of Z-score transformed data following log transformation of the peak areas for the labeled and unlabeled metabolites plotted using GraphPad Prism 10 software.

## Seahorse analysis of cellular respiration and extracellular acidification

GBM12, GBM28, U87, and U87-mIDH cells were seeded in PLL-coated XFe24 cell culture microplates (Agilent Technologies) at $8 \times 10^4$ cells per well in 200 µL culture media ($n \geq 3$ replicates) and were allowed to attach overnight. 24 h post-treatment with oHSV, plates were spun down for 2 min at 400 rcf at room temperature. Media was replaced with 180 µL/well Seahorse Basal media pH7.4 (DMEM, 5 mM HEPES, 2 mM Glutamine, 1 mM Pyruvate, and 10 mM Glucose) and incubated at 37 °C incubator (CO2- less) for ~45 min. Parameters for the Mitostress test were: 1.5 µM oligomycin-A, 2 µM trifluoromethoxy carbonylcyanide phenylhydrazone (FCCP), and 0.5 µM rotenone/antimycin A. OCR and extracellular acidification rate (ECAR) were analyzed using the XFe 96 Flux Analyzer (Agilent, Santa Clara, CA).

## qRT-PCR

Qiagen RNeasy Mini Kit (74104, Qiagen, MD, USA) was utilized for isolating RNA from GBM cells 48 hpi (MOI = 0.01 for GBM12 and 0.03 for GBM28) as per the manufacturer's protocol. cDNA was synthesized from 2.5 µg of quantified RNA using the High-Capacity cDNA Reverse Transcription Kit (Cat. No. 4368814, ThermoFisher Scientific, US). Primers were designed using Primer3 software or obtained from literature and synthesized from Integrated DNA Technologies, Inc. (IDT,

Iowa, US). qPCR was performed using Fast SYBR™ Green Master Mix (Cat. No. 4385612, Fisher Scientific, US) in QuantStudio™ 3 Real-Time PCR System (Applied Biosystems, US) in 96-well plate with the following cycling conditions: initial denaturation 95 °C–2 min; amplification cycle: denaturation 95 °C–15 s, annealing 60 °C–20 s and extension 72 °C–5 s. GAPDH was used as housekeeping control and fold change in expression between groups was calculated using the $2^{(-\Delta\Delta Ct)}$ method. The primers used for the measurement were as follows: human ACSL4 forward: GCTTCCTATCTGATTACCAGTGTTGA; human ACSL4 reverse: GTCCACATAAATGATATGTTTAACACAACT; human LPCAT3 forward: GCGGCTGATCATCTCCATCTT; human LPCAT3 reverse: TTGCCGGTGGCAGTGTAATA; human CPT1 forward: GCTGGAGGTGGCTTTGGT; human CPT1 reverse: GCTTGGCGGATGTGGTTC; human GAPDH forward: GGTTTCTA-TAAATTGAGCCCGCA and human GAPDH reverse: ACCAAATCCGTTGACTCCGA.

## Transmission electron microscopy
GBM12 and GBM28 cells were plated at 500,000 cells/well in a 6-well tissue culture-treated plate. The following day cells were treated with PBS or oHSV (MOI 0.01 for GBM12 and MOI 0.03 for GBM28) for 24 h. Media was replaced with the fixation buffer containing 2.5% glutaraldehyde in 0.1 M cacodylate buffer for 2 h at room temperature. Cells were gently scrapped in a fixative solution, pelleted, and washed twice in 0.1 M cacodylate buffer then post-fixed in 1.0% OsO4 (osmium tetroxide) in 0.1 M cacodylate for 1 h. The cells were then washed twice in de-ionized water and then dehydrated through an ascending ethanol series ending in two changes of 100% ethanol and two changes of PPO (propylene oxide). The cells were infiltrated with a 1:1 mixture of PPO and Eponate 12™ epoxy resin (Ted Pella, Inc.), then 2 changes of 100% epoxy resin, then one last change of 100% fresh epoxy resin. The samples were polymerized over 2 days at 60 °C. The samples were sectioned using Reichert Ultracut S and a Diatome™ diamond knife. The ultrathin (70 to 80 nm) thin sections were collected onto 200 mesh copper grids with carbon-stabilized Formvar™ support film. The sections were then post-stained with 5% Uranyl Acetate (aqueous) and Reynold's lead citrate. The sections were imaged using a JEOL JEM-1400 transmission electron microscope with a LaB$_6$ filament at 80 kV. Images were collected at magnifications of 1000×, 4000×, and 10,000×. Image analysis was done using ImageJ software (NIH, USA).

## FerroOrange staining
48 h after virus treatment, GBM cells were collected by trypsinization. Cells were washed in serum-free media and stained with 1 µM FerroOrange reconstituted in serum-free media at 37 °C for 30 min. Cells were analyzed using a Novocyte Quanteon flow cytometer using PE filter to detect FerroOrange signals. Data was processed using FlowJo v10 software. Cells were gated as described for ROS staining. The intensity of FerroOrange in cells in the gate was visualized in the PE channel and represented as MFI and replicates for each condition were plotted using GraphPad Prism 10 software.

## Cell viability assay
GBM12 cells were seeded in 96-well plates at a density of 20,000 cells/well in 2% FBS-containing media. Cells were infected at MOI = 0.01 with either Q or P followed by treatment with or without CB-839 in 2% FBS-containing media. 48 hpi cell viability was measured using a 2-step MTT assay (Cat. No. 11465007001, Millipore Sigma, USA) as per the manufacturer's protocol. Absorbance was measured using a spectrophotometer (BioTek, USA) and represented as the percentage of live cells compared to uninfected control.

## Immunofluorescence
Immunohistochemistry (IHC) was performed on 4% paraformaldehyde (PFA)-fixed, paraffin-embedded sections. Sections were first deparaffinized at 60 °C for 5–10 min and xylene, followed by rehydration through sequential alcohol gradient, antigen retrieval (10 mM Citrate buffer, pH 6.0 in a boiling water bath), blocking (10% normal goat serum in PBS with 0.5% Tween-20), primary antibody (CD8 (1:500), Cat. No. PA5-88265, ThermoFisher Scientific, USA) incubation, Alexa Flour 488 conjugated F(ab')2-goat anti-rabbit IgG (1:250, Cat. No. A48282 ThermoFisher Scientific, USA) incubation and mounting using ProLong™ Gold Antifade Mountant with DAPI (Cat. No. P36935; ThermoFisher Scientific, USA). Images were acquired on a Nikon Eclipse Ti2 microscope (Nikon, Japan) using a 100× oil objective. Images were exported to ImageJ for further analysis. A minimum of three fields were obtained from each sample (n = 3 mice/group).

## Human PBMCs, T cells, and PBMC-derived Dendritic Cells (DCs)
Human PBMCs were isolated from healthy donors using a buffy coat (Shepeard Community Blood Center, Augusta, Georgia, USA) by Ficoll gradient centrifugation. T-cells were isolated from PBMCs by negative selection using a T-cell isolation kit (Cat. No. 17951, Stemcell Technologies, California, USA). DCs were derived from PBMCs by supplementing the media with 20 ng/mL hGM-CSF (Cat. No. 300-03, PeproTech, USA) and 20 ng/mL hIL-4 (Cat. No. 200-04, PeproTech, USA) for 5 days.

## Flow cytometry
For cell surface staining, cells were washed with phosphate-buffed saline (PBS) and blocked with Fc blocker (Cat. No. 564219, BD Biosciences, California, USA). Fluorochrome-labeled antibodies CD11c (Cat. No. 559877, clone B-ly6), HLA-DR (Cat. No. 562331, clone G46-6), CD8 (Cat. No. 555635, clone HIT8α), and CD69 (Cat. No. 557049, clone FN50) were obtained from BD Biosciences. Cells were stained for 30 min in the dark, washed with PBS, and fixed with 2% PFA before proceeding with analysis. All samples were analyzed on a Novocyte Quanteon flow cytometer (Agilent, California, USA).

## Tetramer staining
Anti-tumor efficacy of oHSV was tested in syngeneic mouse GBM models. 005-OVA cells were intracranially inoculated into female C57BL/6 mice. Tumor-bearing mice were treated with oP10 with or without Ferrostatin-1. Fifteen days post-virus treatment tumor-bearing hemispheres and splenocytes were harvested, dissociated, and homogenized using the tumor dissociation kit (Cat. No. 130-096-730 Miltenyi Biotec, California, USA) as per the manufacturer's instructions[i]. H2Kb-OVA (chicken ova 257-264, NIH Tetramer Core Facility, USA) and H2Kb-gB (HSV-1 gB 498-505, NIH Tetramer Core Facility, USA) tetramers were a kind gift from Dr. Bangxing Hong (Augusta University, GA, USA). Anti-tumor and antiviral immune response were monitored by labeling T cells with CD45 (Cat. No. 560520, BD Biosciences, USA), CD8 (Cat. No. 553031, BD Biosciences, USA) and tetramers against tumor SIINFEKL-H2Kb-OVA and against virus SSIEFARL-H2Kb-gB envelope protein utilizing flow cytometry and plotted as cell count.

## Kinome analysis
Lysates from uninfected and oHSV-infected GBM12 cells were BCA protein and/or dsDNA quantified prior to array loading and analysis utilizing the Tyrosine (PTK) arrays with 15 µg of protein, or Serine/Threonine (STK) arrays using 2ug of protein, as per a standard kinomic protocol. Phosphorylation data was collected over multiple computer-controlled pumping cycles, and exposure times (10–200 ms) for -144 phosphorylatable substrates per array. Raw image analysis was conducted using Evolve2 software, and comparative analysis, upstream kinase prediction, and figure generations were done in BioNavigator v6.2 (PamGene) using scoring predominantly from Kinexus (www.phosphonet.ca). Whole Chip comparative analysis (BioNavigator Upkin PTK v 6.2) was done between groups generating Mean Final

Scores (MFS; combined kinase specificity and sensitivity) scores and Mean Kinase Statistic (MKS; directionality and amount of change). Additionally, network modeling of oHSV-altered kinases was conducted by uploading lists of kinases (MFS > 2.5) by UniProt ID to MetaCore™ (portal.genego.com) and networks were generated using figure-indicated maximum node size ($n < 50$ nodes), with an Auto Expand model, canonical pathways deselected, reactions/metabolites and orphan nodes excluded.

## Lipid peroxidation assay

Cells with appropriate density per 10 cm dish were seeded for 24 h. The following day cells were infected with oHSV followed by treatment of indicated inhibitors/chemicals. Lipid peroxidation was detected by a lipid peroxidation assay kit (Sigma-Aldrich, USA). Briefly, $2 \times 10^6$ cells were homogenized in MDA lysis buffer with butylated hydroxytoluene (BHT). Cleared supernatant was incubated with thiobarbituric acid (TBA) at 95 °C for 1 h to form the colorimetric MDA-TBA adduct. Lipid peroxidation was determined by measuring the absorbance of the colorimetric product at 532 nm. The amount of lipid peroxidation present in the samples was determined from the standard curve and represented relative to uninfected control samples.

## Free fatty acid assay

The free fatty acid was detected by a free fatty acid assay kit (Sigma, USA). Briefly, cell supernatants and palmitic acid standards were collected and put into a 96-well plate (50 μL). Then 2 μL of ACS Reagent was added and incubated, followed by the addition of 50 μL of the Master Reaction Mix. After incubation, the absorbance at 570 nm (A570) was measured. The fatty acid present in the samples was determined from the standard curve.

## DAMPs measurement

**Extracellular ATP (eATP).** $1 \times 10^5$ cells/well were seeded in 24-well plates and allowed to adhere to tissue culture plates overnight. Cells were infected with oHSV for 1 h. The unbound virus was removed, and media was replaced without or with Fer-1. 24 hpi, conditioned media was harvested, and ATP concentration was assessed using an ATP determination kit (Thermo Fisher).

**HMGB1 ELISA.** Culture supernatants from oHSV-infected cells in the presence or absence of Fer-1 were collected and centrifuged at a low speed to get rid of the cell debris. HMGB1 released from these cells was measured by ELISA as per the manufacturer's guidelines (Sigma-Aldrich, USA). The concentration of HMGB1 was represented as pg/mL.

## Metabolite analysis

**Pyruvate dehydrogenase (PDH) assay.** PDH activity was measured using the PDH Activity Assay Kit (Sigma-Aldrich, USA). Briefly, $1 \times 10^6$ cells were homogenized in 100 mL of ice-cold PDH Assay Buffer and clarified supernatant was used for the assay. PDH activity was determined by the amount of NADH generated per reaction time per mL of sample measured at 450 nm.

**Pyruvate kinase (PK) assay.** PK activity was measured using the PK Activity Assay Kit (Sigma-Aldrich, USA). Briefly, $1 \times 10^6$ cells were rapidly homogenized with 4 volumes of PK Assay Buffer and centrifuged to remove insoluble materials. Pyruvate concentration was determined from the absorbance of the colorimetric product at 570 nm, proportional to the pyruvate present.

**Acetyl-CoA assay.** Acetyl-CoA activity was measured using the Acetyl-CoA Assay Kit (Sigma-Aldrich, USA) following the manufacturer's instructions. Samples were deproteinized by perchloric acid precipitation, homogenized, and cleared supernatant neutralized with 3 M potassium bicarbonate solution. Acetyl-CoA concentration was

determined by the resulting fluorometric ($\lambda ex = 535/\lambda em = 587$ nm) product, proportional to the amount of Acetyl-CoA present.

## Western blot

Cultured cells were infected with oHSVs or saline at MOI 0.01 for 48 h. Cells were lysed in RIPA cell lysis buffer with protease and phosphatase inhibitors added prior to use. Lysed cells were sonicated, and protein concentration was measured by BCA and 40 μg protein was loaded in 4%–20% pre-cast gels. Proteins were transferred to PVDF membranes and subsequently probed with primary antibodies overnight at 4 °C. Primary antibodies used were: ACSL4 (Cat. No. MA5-31548, Thermo-Fisher Scientific, USA), 4-HNE (Cat. No. MA5-27570, ThermoFisher Scientific, USA), GPX4 (Cat. No. 52455S, Cell Signaling Technology, USA), CPT1A (Cat. No. 12252S, Cell Signaling Technology, USA), LPCAT3 (Cat. No. 72964, Cell Signaling Technology, USA) and pan-p-PKC (betaII S660) (Cat. No. 9371, Cell Signaling Technology, USA). Membranes were washed with TBS + 0.05% Tween-20 (TBST) three times and then incubated with HRP-conjugated secondary antibodies (goat anti-rabbit IgG (Cat. No. 7074, Cell Signaling Technology, USA) and horse anti-mouse IgG (Cat. No. 7076, Cell Signaling Technology, USA)) for 1 h at room temperature (RT). Membranes were developed using the ChemiDoc MP Imaging System (Bio-Rad, USA). Anti-Beta tubulin antibody (Cat. No. 2128L, Cell Signaling Technology, USA) was used as both protein transfer and loading control.

## Protein Kinase C (PKC) activity assay

Equal amounts of protein were used for the measurement of total kinase activity using a PKC kinase activity kit from Enzo (Cat. No. ADI-EKS-420A, Enzo Life Sciences, USA) following the manufacturer's instructions. Briefly, $2 \times 10^6$ cells were treated ± oHSV (Q or P) followed by ± inhibitors in 10 cm tissue culture-treated dishes. 48 h post-treatment, cells were lysed in RIPA lysis buffer. 30 μL of lysate per sample was used to measure PKC kinase activity at 450 nm wavelength using a spectrophotometer (BioTek, USA). Kinase activity was normalized by protein concentration and represented as relative PKC activity compared to untreated control.

## Diacylglycerol (DAG) assay

Quantification of DAG was done using the DAG assay kit (Cat. No. ab242293, Abcam, USA) according to the manufacturer's instructions. Briefly, 48 h post-treatment $10^7$ cells/group were homogenized in chilled 1× PBS buffer, sonicated, centrifuged and the DAG obtained after drying the organic phase using a speedvac was resuspended in 1× assay buffer. 20 μL of the sample was used to quantify the amount of DAG using a fluorescence microplate reader (BioTek, USA) at Excitation/Emission = 530–560 nm/585–595 nm range, represented as μM.

## Magnetic resonance imaging (MRI)

MRI scans were carried out on a 7 T Bruker Biospec 70/20 (Bruker Biospin, Billerica, MA) equipped with an 86 mm quadrature transmit coil and a dedicated 4-element phased array coil. The animals were anesthetized with a mixture of isoflurane/medical air (3% for induction and 1%–2% for maintenance) via a nose cone. The mice were placed in a prone position on a dedicated mouse bed with a circulating warm-water circuit to maintain body temperature. Respiration rate and rectal temperature were continuously monitored through the experiments (SA-instruments, Stony Brook, NY). MRI contrast agent, Gadolinium diethylenetriaminepentaacetic acid (Gd-DTPA), was administrated i.v. at a dose of 0.2 mM/kg to obtain signal enhancement in the tumor. Multi-slice T1-weighted spin echo images were obtained in the coronal orientation using a repetition time of 1500 ms, echo time of 8 ms, and imaging matrix of 256 × 256 with the field of view of 19.2 × 19.2 mm². To match the histological analysis, a slice thickness of 0.75 mm was used without a slice gap. The number of signal average was 4 for all the scans. T1-weighted spin echo imaging was done before and after

administration of the contrast agent for each animal using the same imaging parameters.

## Statistical analysis

Statistical analysis and generation of graphs were performed using GraphPad Prism v10.0. *P* values were calculated using two-way analysis of variance (ANOVA) followed by Sidak's correction for multiple comparisons of cell means to every other cell mean in that row; one-way ANOVA followed by Tukey's post hoc test for multiple comparisons using statistical hypothesis testing; unpaired two-tailed Student t-test; one-tailed Student t-test with unequal variances; Mann Whitney U test where appropriate. A *p* value of 0.05 was the cutoff for statistical significance. Log-rank (Mantel–Cox) test was used for the analysis of survival studies in the mouse model. Data are reported as mean ± standard deviation (S.D.) except Fig. 3a, b, e which is mean ± standard error of the mean (SEM) from $n \geq 3$ independent replicates. The number of replicates and details of the statistical tests employed are also described in the figure legends.

## Reporting summary

Further information on research design is available in the Nature Portfolio Reporting Summary linked to this article.

## Data availability

Raw and processed single-cell RNA-Seq data from mouse tumors are available at GEO Hub with accession number GSE289317. RNA-Seq data from Clinical Trials NCT00028158[7] and NCT03152318[8] were obtained from publicly available data repositories. Uncropped scans of all blots in Figures and Supplementary Figures are provided in the Source Data file. The remaining data are available within the Article, Supplementary Information or Source Data file. Source data are provided with this paper.

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

## Acknowledgements

This study was supported by grants from NIH/NCI P01CA163205 (to B.K.), NIH/NINDS R01NS127473 (to BK), NIH/NINDS R61NS112410 (to B.K.), CPRIT Proteomics and Metabolomics Core Facility RP210227 (to N.P.), NIH R01CA282282 (to N.P.), NIH P30 CA125123 (to N.P.), Dan L. Duncan Cancer Center (to N.P.) and Georgia Cancer Center Paceline Trainee Research Award 8470T (to U.S.). Dr. Maria Castro (University of Michigan, MI, USA) provided murine wtIDH1 and mutIDH1[R132H] cells. We acknowledge Robert P. Apkarian Integrated Electron Microscopy Core at Emory University and Augusta University Electron Microscopy and Histology Core for Transmission of electron microscopic images. We wish to acknowledge M. Zoccheddu, M. Yu, and S.S. Manam of Integrated Genomics Shared Resources at Georgia Cancer Center at Augusta University. We thank Dr. A. Bosomtwi and Dr. R. Ara from Small Animal Imagin Shared Resources at Georgia Cancer Center at Augusta University for conducting the MRI. We created all the schematics using BioRender.com.

## Author contributions

Conceptualization and design: U.S., M.P.M., and B.K. Methodology: U.S., M.P.M., S.A.M., J.C.A., V.P., A.H.M.K., P.W., and J.H.P. Analysis: U.S., M.P.M., J.C.A., C.W., V.P., N.P., P.W., A.L.L., and B.K. AS and TJL performed RNA-seq and scRNA-seq analysis. Clinical data: E.A.C. and J.M.M. Writing original draft: U.S. and B.K. All authors edited and reviewed the manuscript.

## Competing interests

B.K. is an inventor of oHSV-P10 (oP10) which has been licensed to Mesoblast. B.K. does not own any financial shares or interest in Mesoblast. E.A.C. is currently an advisor to Bionaut Labs, Insightec, Inc., Seneca Therapeutics, and Theriva. He has equity options in Bionaut Laboratories, Seneca Therapeutics, and Ternalys Therapeutics. He is a co-founder and on the Board of Directors of Ternalys Therapeutics. He has received research support from NIH, US Department of Defense, American Brain Tumor Association, National Brain Tumor Society, Alliance for Cancer Gene Therapy, Neurosurgical Research Education Foundation, Advantagene, NewLink Genetics, and Amgen. He also is a named inventor on patents related to oncolytic HSV-1 and noncoding RNAs. J.M.M. is a board and equity holding member, in Aettis, Inc., and may receive royalties. The company holds frozen oncolytic viral stocks. Mustang Bio-Tech is licensing the Intellectual Property (IP) of C134, an oncolytic viral Therapy. J.M.M. is blinded to the conditions for the C134 clinical trials. He is a shareholder for a privately held Small Business Innovation Research LLC, Treovir, Inc., concerning G207 oncolytic viral therapy now in clinical trial. Merck, Inc. provides industry grant support by providing Keytruda (pembrolizumab) for a clinical trial of M032 oncolytic virotherapy and financial support for a clinical trial. J.M.M. is a listee on Intellectual Property: 1. related to a cancer immunotherapy system, and 2. to a novel immuno-virotherapeutic strategy targeting the glioma secretome. This IP has been filed by in8Bio (formerly Incysus, Ltd.) and has royalty-earning potential. All other authors declare no competing interests.
