## [Transparent Peer Review file · Nature Communications]

IDH status dictates oHSV mediated metabolic reprogramming affecting anti-tumor immunity.

Corresponding Author: Professor Balveen Kaur

Version 0:

Reviewer comments:

Reviewer #1

(Remarks to the Author)

In the field of oncolytic virotherapy (OV), understanding how OV can efficiently induce antitumor immunity is crucial. This study by Sahu et al. identified that isocitrate dehydrogenases (IDH) play an important role in modulating antitumor immunity in the context of oncolytic herpes simplex virus (oHSV)-mediated virotherapy by controlling metabolic reprogramming. Specifically, the authors observed an increased reactive oxygen species (ROS), reduced glutathione, and induction of acyl-CoA synthetase long-chain family member 4 (ACSL4) gene expression during oHSV virotherapy in glioma cells with wild-type IDH. They found that, together, these factors resulted in lipid peroxidation and ferroptosis, which promoted antitumor immunity. However, as IDH is frequently mutated in glioma cells, it creates an unfavorable metabolic microenvironment for generating antitumor immunity during oHSV virotherapy. The authors conducted experiments showing that pharmacological blockade of mutated IDH function in glioma cells could induce ferroptosis and enhance antitumor immunity. Overall, this is a well-designed study with sufficient data to support its major conclusions, effectively combining mechanistic insights with translational exploration.

Major points

1. The data in Fig. 1 showed that enrichment of GSEA pathways related to oxidative phosphorylation, mitochondrial respiration, and ATP synthesis were only detected in HSV seropositive patients but not in HSV seronegative patients. Since oHSV was administered to patients intratumorally, it is assumed that the virus should infect tumor cells at almost equal efficiency; it is wondered why there were such significant differences in these parameters. The authors need to provide some explanation, this is despite the observation in the cited clinical studies that the antitumor immunity and the therapeutic outcome between these two patient groups are different.
2. For cells with IDH mutation, e.g., IDH1R132H, do these cells also contain a copy of wild-type IDH? If not, once mutIDH is blocked with a blocking agent (e.g., AGI-519), there will be no IDH activity one way or the other. So it will not receive the impacts of wild type IDH. Then, where do the therapeutic benefits (e.g., those seen in Fig.8) come from?
3. How specific is AGI-519? It was used in Fig. 8 for both human and mouse tumors. Does it have the capacity to inhibit both human and mouse mutIDH? Does it inhibit wild type IDH as well?

Minor points

1. Inconsistent usage of cells with IDH mutation, including IDH1R132H, IDH132H, and mutIDH, very confusing.

Reviewer #2

(Remarks to the Author)

This manuscript by Sahu et al., reports elegant and timely data related to the impact of IDH status in impacting oHSV mediated metabolic reprogramming which in turn affects anti-tumor immunity. The work is of high quality and the results and experimental designs are rigorous. The conclusions, for the most part, are also in line with the experimental data presented.

Please find several comments enclosed below:

- 1- The authors present data from a recently completed clinical trial using oHSV, in GBM patients. They should provide a clearer and more compelling rationale of why they move from WT IDH GBM in human patients to work performed in periclinal mouse models of mIDH1 glioma. As it stands the mIDH1 section of this manuscript is disjointed from the main body of the work reported.
- 2- It would be informative if the authors could present heat maps related to their RNA-seq data as reported in Figure 1
- 3- The authors should comment on the biological relevance of the succinic acid data presented in Figure 2 H, especially taking into account the extremely low levels reported.
- 4- The data shown in Figure 3 should be reproduced in at least one other WT IDH GBM cell culture obtained from surgical biopsies and to validate the data related to mutant IDH, they should also perform these experiments utilizing mIDH human cells in culture. There are several mIDH huma glioma cells available.
- 5- Same comments apply for the data shown in Figure 4, Figure 5 and Figure 6. More than one GBM cell culture should be used to validate this data. Specially taking into account the highly heterogeneous nature for GBMs.
- 6- The immunological read outs of this work need to be strengthened. The authors should consider performing tumor specific T cell proliferation assays and cytolytic activity assays form their in vivo experiments.

Reviewer #3

(Remarks to the Author)

In the current study, the authors investigate the role of lipid metabolism and ferroptosis in mediating oncolytic HSC-induced tumor suppression. Transcriptomic analysis revealed a significant enrichment of pathways that drive oxidative phosphorylation. Metabolic analysis including glucose carbon tracing, glutamine flux and seahorse mito stress test found an increase in glucose utilization, reductive carboxylation of glutamine and oxidative phosphorylation which subsequently resulting in elevated reactive oxygen species (ROS) and acyl-CoA level. Besides, they found PKC was involved in the ACSL-mediated lipid metabolism and ferroptosis induction, while inhibiting PKC or glutamine metabolism blocked ferroptosis caused by virotherapy, suggesting the indispensable role of ferroptosis in the efficacy of virotherapy. Though these findings imply a potential function of ferroptosis in virotherapy for cancer treatment, several questions should be thoroughly addressed before consideration for publication.

- 1, In Fig.1, the authors analyzed the samples after virotherapy to find the metabolic features, which guides them to follow this metabolic study. However, it's known that virus might enhance the metabolic ability in most cases, which could be one reason during tumor initiation upon virus infection. There should be several controls to show whether this metabolic feature is unique to oHSC-mediated virotherapy or not.
- 2, In Fig4, the study used Nile red assay to indicate increased lipid metabolism, which was too preliminary to conclude the regulation of lipid metabolism by oHSC. Lipidomic analysis should be used to confirm specific lipids in the relevant pathways are involved.
- 3, In Fig.5A, the authors should measure different ROS with specific dyes, including mito ROS, since mito function is also impaired in the context.
- 4, In Fig.5B, what does free GSH mean? Either reduced GSH or GSSG level is fine, the ratio of GSH and GSSG is the ideal choice.
- 5, The logic to focus on ferroptosis in Fig.5 is not clear to the reviewer. There are many type of cell death involved ROS and lipid metabolism, what's the key clue of directing the authors to study ferroptosis is not convincing.
- 6, Why do the authors use kinase screening assay to identify the pathway invovled, is it known that oHSV could increase phosphorylation of many proteins? Whether the transcriptomic analysis indicates downstream changes of specific kinase like PKC?
- 6, The role of PKC and/or ACSL4 in mediating oHSV-induced ferroptosis is not demonstrated in the context.
- 7, The protein level of target of PCK should be verified, such as ACSL4 and LPCAT3.
- 8, In Fig.5, the data couldn't demonstrate the occurrence of ferroptosis, since ferroptosis inhibitor wat not employed at this stage yet.
- 9, Glutaminase promotes the generation of glutamate, which could be also used for export in exchange of cystine uptake, which is key for ferroptosis inhibition. Thus, the effect of GLS inhibitor on ferroptosis regulation might involve cystine metabolism, which should also be considered and studied.
- 10, The study didn't answer the core question of why oHSC could enhance glucose and glutamine metabolism.
- 11, It's interesting to figure out the exact of ferroptosis in tumor cells and ferroptosis-mediated immune activation respectively in oHSC-induced tumor suppression, in other words , which one is dominant in terms of killing tumor cells, ferroptosis in tumor cells or ferroptosis-mediated immune activation?

Version 1:

Reviewer comments:

Reviewer #1

(Remarks to the Author)

The authors have addressed my critiques/concerns to satisfaction. In my opinion, the manuscript is ready for publication in this journal.

Reviewer #2

(Remarks to the Author)

This revised manuscript by Sahu et al, is highly significant for the field of oncolytic virotherapy and neuro-oncology. The results are timely, rigorous, and the conclusions are in line with the data presented by the authors.

Please find enclosed below minor comments:

1- In the materials and methods section, the authors should cite the paper by Nunez F. J. et al Sci Transl Med. 2019 Feb 13;11(479):eaaq1427. doi: 10.1126/scitranslmed.aaq1427. PMID: 30760578

2- If available, they should also add the relevant reference for the human cells utilized.

3- The authors should also add a sentence related to the fact that it would be valuable to uncover the mechanism by which oHSV manipulates cellular metabolic pathways, as one of the limitations of their study. Understandably, I do agree with the authors, that this is beyond the scope of the current study.

Reviewer #3

(Remarks to the Author)

Several ferroptosis-related markers, such as mitochondria condensation, lipid peroxidation, and alteration of several proteins, are involved in ferroptosis regulation. However, none of them exclusively indicate the occurrence of ferroptosis, simply because at these stages cells might not be undergoing death yet. The reliable way to demonstrate ferroptotic cell death is the results of cell death suppression by ferroptosis inhibitors.

Reviewer #1 (Remarks to the Author):

In the field of oncolytic virotherapy (OV), understanding how OV can efficiently induce antitumor immunity is crucial. This study by Sahu et al. identified that isocitrate dehydrogenases (IDH) play an important role in modulating antitumor immunity in the context of oncolytic herpes simplex virus (oHSV)-mediated virotherapy by controlling metabolic reprogramming. Specifically, the authors observed an increased reactive oxygen species (ROS), reduced glutathione, and induction of acyl-CoA synthetase long-chain family member 4 (ACSL4) gene expression during oHSV virotherapy in glioma cells with wild-type IDH. They found that, together, these factors resulted in lipid peroxidation and ferroptosis, which promoted antitumor immunity. However, as IDH is frequently mutated in glioma cells, it creates an unfavorable metabolic microenvironment for generating antitumor immunity during oHSV virotherapy. The authors conducted experiments showing that pharmacological blockade of mutated IDH function in glioma cells could induce ferroptosis and enhance antitumor immunity. Overall, this is a well-designed study with sufficient data to support its major conclusions, effectively combining mechanistic insights with translational exploration.

Major points

1. The data in Fig. 1 showed that enrichment of GSEA pathways related to oxidative phosphorylation, mitochondrial respiration, and ATP synthesis were only detected in HSV seropositive patients but not in HSV seronegative patients. Since oHSV was administered to patients intratumorally, it is assumed that the virus should infect tumor cells at almost equal efficiency; it is wondered why there were such significant differences in these parameters. *The authors need to provide some explanation, this is despite the observation in the cited clinical studies that the antitumor immunity and the therapeutic outcome between these two patient groups are different.*

Response: We agree with the reviewer that while all patients were inoculated with oHSV, some did not convert to seropositivity after virus treatment. The reasons why these patients did not develop an antibody response despite virus inoculation is indeed intriguing.

We have added the following statement in the discussion of the revised manuscript (Lines 808-812):

“It is interesting to note that some patients did not convert to become HSV-1 seropositive even after virus treatment, and that these patients were associated with significantly worse response to treatment [20]. While the reason why these patients did not convert is not clear it can be speculated that this might reflect a highly immune suppressed state of these patients.”

2. For cells with IDH mutation, e.g., IDHR132H, do these cells also contain a copy of wild-type IDH? If not, once mutant IDH is blocked with a blocking agent (e.g., AGI-519), there will be no IDH activity one way or the other. So it will not receive the impacts of wild type IDH. Then, where do the therapeutic benefits (e.g., those seen in Fig.8) come from?

Response: wtIDH is an enzyme that catalyzes the reversible conversion of isocitrate to α -KG thus maintaining levels of α -KG in a cell. Mutant IDH (R132 and R172) represent a gain of function, neomorphic mutation that imparts the gene a new function. Mutant IDH catalyzes the utilization of α -KG to form 2-HG, hence blocking reductive carboxylation (PMID: 22442146, PMID: 24755473). Since this imparts a dominant function, a single mutant allele in tumors can drive the phenotype and thus mutant

IDH is frequently observed as a heterozygous mutation in cancers (PMID: 20171147). The mutant IDH mouse model also retains wtIDH to mimic the human heterozygosity. Thus, glioma cells with mutant IDH also harbor a copy of wtIDH. When these cells are treated with AGI-5198, mutant IDH loses its function, and wtIDH in these cells can restore the balance of α -KG to permit reductive carboxylation.

We have now clarified in the manuscript that these cells are heterozygous (in the results section “**Effect of mutant IDH on virus efficacy**”). We have added the following to clarify (Lines 719-726):

“The IDHR132H neomorphic mutation confers a gain-of-function activity resulting in the reduction of α -KG to the oncometabolite 2-hydroxyglutarate (2-HG) (Fig. 9I) [35, 36]. Thus, this frequently occurs as a heterozygous mutation in glioma patients. The cell lines used here also retain this heterozygosity and thus when IDHR132H glioma cells are treated with AGI-5198 (IDHR132H inhibitor), wtIDH allele can restore the balance of α -KG to permit reductive carboxylation and lead to enhanced lipid peroxidation (Fig. 9J) [37].”

3. How specific is AGI-5198? It was used in Fig. 8 for both human and mouse tumors. Does it have the capacity to inhibit both human and mouse mutant IDH? Does it inhibit wild type IDH as well?

Response: AGI-5198 is a mutant IDH1 inhibitor that reduces the production of 2-HG in IDH1 mutant human and murine cells (PMID: 23558169; PMID: 28986582). This drug forms the basis for Ivosidenib, an FDA approved mutant IDH inhibitor for the treatment of AML (PMID: 36091829). While we cannot rule out off target effects for any drug, to my knowledge this drug’s effect on other cellular targets has not been reported (PMID: 23558169).

Minor points

1. Inconsistent usage of cells with IDH mutation, including IDH1R132H, IDHR132H, and mutant IDH, very confusing.

Response: We apologize. We have gone through and used mutant IDH or IDHR132H as applicable throughout.

Reviewer #2 (Remarks to the Author)

This manuscript by Sahu et al., reports elegant and timely data related to the impact of IDH status in impacting oHSV mediated metabolic reprogramming which in turn affects anti-tumor immunity. The work is of high quality and the results and experimental designs are rigorous. The conclusions, for the most part, are also in line with the experimental data presented.

Please find several comments enclosed below:

1- The authors present data from a recently completed clinical trial using oHSV, in GBM patients. They should provide a clearer and more compelling rationale of why they move from WT IDH GBM in human patients to work performed in periclinical mouse models of mutant IDH1 glioma. As it stands the mutant IDH1 section of this manuscript is disjointed from the main body of the work reported.

Response: We have added the rationale for these experiments in the beginning of the results section “Effect of mutant IDH on virus efficacy”. We have added the following statement in **lines 706-715**:

“Collectively, analysis of the transcriptomic data from patient tumor specimens after virotherapy and our in vitro and in vivo analysis shows that oHSV treatment drives glucose utilization into the TCA cycle and glutamine metabolism towards reductive carboxylation to increase fatty acid synthesis. Changes in cellular metabolism have an impact on cancer growth and in particular mutations in IDH enzyme dramatically affect glioma biology and prognosis, and cells harboring mutant IDH cells cannot undergo reductive carboxylation [9, 34]. Interestingly in NCT03152318 trial, patients with wt IDH had a better response to oHSV than patients harboring mutant IDH. Thus, to evaluate the significance of oHSV treatment on mutant IDH glioma we compared matched murine glioma cells harboring heterozygous mutant IDHR132H (IDHR132H) to wildtype IDH (wtIDH) glioma [9].”

2- It would be informative if the authors could present heat maps related to their RNA-seq data as reported in Figure 1

Response: We have added heatmaps to the Supplemental Figure 1 (**Fig. S1A-C**).

3- The authors should comment on the biological relevance of the succinic acid data presented in Figure 2 H, especially taking into account the extremely low levels reported.

Response: The low levels of U-13C labeled succinic acid compared to unlabeled succinate could be attributed to high endogenous concentrations from other sources. We specifically calculated the percentage of incorporation only from labeled glucose. The raw data is provided in the source data for transparency.

4- The data shown in Figure 3 should be reproduced in at least one other WT IDH GBM cell culture obtained from surgical biopsies and to validate the data related to mutant IDH, they should also perform these experiments utilizing mIDH human cells in culture. There are several mIDH huma glioma cells available.

Response: Thank you for the suggestion. In the revised manuscript, we have now performed oxygen consumption analysis on multiple glioma cell lines. Oxygen consumption data on human wtIDH GBM cells (GBM28), is added in the revised Figure 3B and shows increased oxygen consumption in cells treated with oHSV.

We have now also repeated sea horse analysis for matched U87 wt and mIDH bearing cells (Fig 9). Revised Fig 9 A-B shows increased oxygen consumption after oHSV treatment in wt U87 glioma cells but not in IDHR132H bearing U87 glioma cells.

5- Same comments apply for the data shown in Figure 4, Figure 5 and Figure 6. More than one GBM cell culture should be used to validate this data. Specially taking into account the highly heterogeneous nature for GBMs.

Response: We have now included data for two different human GBM cells in all three revised figures.

In the revised Figure 4, we now show increased Nile red staining (Fig 4F-G), lipid droplets by TEM (Fig 4H-I), and increased relative fatty acid content (Fig 4J-K) after oHSV therapy in GBM12 and GBM28 glioma cells.

In revised Figure 5, we now show increased ROS, reduction in the amount of reduced glutathione (GSH), increased PKC activity, increased ACSL4, increased lipid peroxidation etc. in both GBM12 and GBM28 glioma cells.

In revised Figure 6, we now show that oHSV induced ferroptosis is not observed in cells treated with inhibitors of electron transport chain, ROS scavengers, and PKC and glutaminase inhibitors in both GBM12 and GBM28 glioma cells.

6- The immunological read outs of this work need to be strengthened. The authors should consider performing tumor specific T cell proliferation assays and cytolytic activity assays from their in vivo experiments.

Response: Thank you for this suggestion. We agree it is important to understand the immunological changes in tumors after virus treatment that can be attributed to ferroptosis. Thus, we have repeated in vivo studies in mice treated with virus and virus plus ferrostatin-1 (Fer-1, to block ferroptosis) to evaluate the importance of ferroptosis in harnessing antitumor immune responses in vivo (new Figure 8 in the revised manuscript). A summary of this new results shown in new Figure 8 are as below:

We performed unbiased single cell sequencing of CD45+ve cells isolated from 005 glioma bearing murine brain hemispheres 5 days after oHSV treatment with or without Fer-1 treatment. There was not an obvious difference between immune cell types recruited to tumors with virus treatment with or without Fer-1 (Figure. 8A). However, Cell chat analysis revealed that both number and significance of interactions from T cells and macrophages were significantly altered in mice treated with Fer-1(Figure. 8B-C). A sub-clustering of T cells showed that Fer-1 treatment reduced the number of Gzmk+ve cytotoxic T cells, and gamma delta T cells. This was accompanied by an increase in exhausted T cells, and Treg cells indicating that ferroptosis was important for T cell activation after virotherapy (Figure. 8E). To evaluate the importance of ferroptosis in T cell functionality (Figure. 8G-H) we evaluated the number of tumor and virus antigen-specific CD8 T cells as described (PMID: 36796878). Briefly, mice bearing 005-OVA glioma were treated with oP10 (P) ± Fer-1. 15 days post virus treatment mice were sacrificed and tumor bearing hemispheres were analyzed for OVA and gB recognizing T cells by tetramer analysis. Flow cytometry revealed a significant increase in OVA+/CD8+ T cell count with oP10 treatment which was inhibited by blocking ferroptosis with Fer-1 treatment (Figure. 8G). Interestingly, there was no significant difference in antiviral gB recognizing T cells (Figure 8H). Collectively this shows that ferroptosis is important for activation of antitumor T cells after virus treatment.

Reviewer #3 (Remarks to the Author)

In the current study, the authors investigate the role of lipid metabolism and ferroptosis in mediating oncolytic HSC-induced tumor suppression. Transcriptomic analysis revealed a significant enrichment of pathways that drive oxidative phosphorylation. Metabolic analysis including glucose carbon tracing, glutamine flux and seahorse mito stress test found an increase in glucose utilization, reductive

carboxylation of glutamine and oxidative phosphorylation which subsequently resulting in elevated reactive oxygen species (ROS) and acyl-CoA level. Besides, they found PKC was involved in the ACSL-mediated lipid metabolism and ferroptosis induction, while inhibiting PKC or glutamine metabolism blocked ferroptosis caused by virotherapy, suggesting the indispensable role of ferroptosis in the efficacy of virotherapy. Though these findings imply an potential function of ferroptosis in virotherapy for cancer treatment, several questions should be thoroughly addressed before consideration for publication.

1, In Fig.1, the authors analyzed the samples after virotherapy to find the metabolic features, which guides them to follow this metabolic study. However, it's known that virus might enhance the metabolic ability in most cases, which could be one reason during tumor initiation upon virus infection. There should be several controls to show whether this metabolic feature is unique to oHSV-mediated virotherapy or not.

Response: Figure 1 is a retrospective analysis of transcriptome data from high grade brain tumor patients enlisted on two phase I clinical trials with oHSV. We do not have access to tissues from patients before and after infection with w.t HSV-1 to examine how w.t virus affects metabolism in normal brain tissue. Nevertheless, our data shows that oHSV treatment in vitro and in vivo affects tumor metabolism. While analysis of how different viruses perturb the metabolic changes in different cell and tissue types is a very important question, we believe it is beyond the scope of this current manuscript.

2, In Fig4, the study used Nile red assay to indicate increased lipid metabolism, which was too preliminary to conclude the regulation of lipid metabolism by oHSV. Lipidomic analysis should be used to confirm specific lipids in the relevant pathways are involved.

Response:

The question of specific lipids that are changes is very interesting. We have now measured fatty acids (FAs) and found alterations in FAs that serve as precursors for lipid synthesis (Figure 1 below, for reviewers only). The data shows a very significant increase in several groups of fatty acids in glioma cells after oHSV treatment. This brings up several interesting questions about the impact of this change on tumors and tumor microenvironment. We plan to pursue this in a separate study and believe that is beyond the scope of this current manuscript.

Nevertheless, we have solidified our observations of increased fatty acid synthesis in 2 different GBM cells in this revised manuscript. In this revised manuscript we observed increased acetyl CoA levels in oHSV treated cells. WE also show increased Nile red staining in two different GBM cells after oHSV treatment. TEM images of 2 different GBM cells also show increased lipid droplets and also increased fatty acid synthesis after virotherapy. We have now performed GSEA analysis from RNA-seq data, which indicates the involvement of lipid metabolism. This is included in the revised Supplementary Figure. S4K. Thus, our data shows that there is increased fatty acids in GBM cells treated with oHSV.

[Figure redacted]

Figure 1 for reviewers only: [Redacted]

3, In Fig.5A, the authors should measure different ROS with specific dyes, including mito ROS, since mito function is also impaired in the context.

Response: Thank you for suggestion. We have now performed the requested staining with MitoSOX in GBM12 and U87 cells shown in revised Figure 5B, and Figure 9G-H.

4, In Fig.5B, what does free GSH mean? Either reduced GSH or GSSG level is fine, the ratio of GSH and GSSG is the ideal choice.

Response: Thank you for the correction. We have now edited the manuscript to specify reduced glutathione (GSH).

5, The logic to focus on ferroptosis in Fig.5 is not clear to the reviewer. There are many type of cell death involved ROS and lipid metabolism, what's the key clue of directing the authors to study ferroptosis is not convincing.

Response: Ferroptosis is an iron-dependent form of cell death that is distinct from apoptosis, necrosis, and autophagy etc. Death ensues due to excessive ROS mediated lipid peroxidation. Apart from oxidative phosphorylation (Figure 1) fatty acid metabolism and ferroptosis related GSEA pathways were significantly enriched in patient tumors after oHSV treatment (Figure S1E). Thus, we evaluated changes in cellular glucose and glutamine metabolism in an unbiased manner using uniformly C13 labeled glucose and glutamine. Our results showed that oHSV treatment resulted in increased lipid synthesis, increased

ROS and increased lipid peroxidation. TEM images of infected cells also show hallmarks of ferroptosis. Thus, we further investigated the implications of ferroptosis on oHSV immune therapy (new Figure 8).

6, Why do the authors use kinase screening assay to identify the pathway involved, is it known that oHSV could increase phosphorylation of many proteins? Whether the transcriptomic analysis indicates downstream changes of specific kinase like PKC?

Response: We agree, oHSV induces and suppresses several different signaling pathways. Here we utilized pan kinome analysis to evaluate in an unbiased manner changes in activity of cellular kinases upon infection with two different oHSVs. We utilized Pam gene technology which facilitates real-time measurement and understanding of alterations in kinase activities. The multiplex properties of the chip enable us to look at the bigger picture of cellular signaling. It permits evaluation of alterations in the activity of multiple pathways. To our knowledge this has not been utilized to evaluate how oHSV affect tumor cell signaling.

6, The role of PKC and/or ACSL4 in mediating oHSV-induced ferroptosis is not demonstrated in the context.

Response: To test the importance of PKC activation on lipid peroxidation we evaluated the effect of inhibiting PKC activity on lipid peroxidation (Fig 6J-K), induction of ACSL4 and HNE (Fig 6 L). Our results showed that oHSV mediated lipid peroxidation and HNE was abrogated when PKC was inhibited. In vivo, treatment of glioma bearing mice with PKC pathway inhibitor Enzasturin (Enza) blocked oHSV induced therapeutic benefit in immune competent mice (Figure. 7L) and inhibited oHSV treatment induced increased CD8+ T cell influx into the mice brain tumors (Figure. 7M-N). Thus, our data shows that PKC activity is important for inducing oHSV induced ferroptosis and antitumor immune responses in glioma cells.

7, The protein level of target of PCK should be verified, such as ACSL4 and LPCAT3.

Response: Thank you for the suggestion.

We have now added the protein level of ACSL4 in uninfected and oHSV infected cell lysates from GBM12 and GBM28 glioma cells (Figure. 5I). LPCAT3 protein levels were induced in GBM12 cells after oHSV (Q and P) treatment. This data is now included as **supplementary Figure S5H**.

8, In Fig.5, the data couldn't demonstrate the occurrence of ferroptosis, since ferroptosis inhibitor was not employed at this stage yet.

Response: Figure 5 shows that oHSV infection leads to an increase in cellular and mitochondrial ROS with a concurrent decrease in reduced glutathione in glioma cells (Figure 5A-E). oHSV infected glioma cells also had increased ACSL4 protein levels (required for the production or incorporation of activated PUFAs into membrane phospholipids) and reduced GPX4 (for the reduction of lipid hydroperoxides) resulting in increased lipid peroxidation and increased FerroOrange +ve cells (Figure 5I and Figure 5K-N) all of which are indicative/markers of ferroptosis. TEM images from two different glioma cells infected with oHSV also showed increased number and reduced mitochondria size which are considered as the morphological hallmarks of ferroptosis (PMID: 33553189, PMID: 24844246) (Figure 5O-R). All of these

together show all the hallmarks of ferroptosis, hence we concluded that oHSV treatment induced ferroptosis: (<https://doi.org/10.1146/annurev-cancerbio-030518-055844>, PMID: 35803244).

Figures 6 and Figures 7 show the implications of ferroptosis on immune response to virotherapy. Our data show that inhibition of ferroptosis or any of the metabolic changes that result in oHSV induced Ferroptosis reduce benefit from oHSV immunotherapy in vivo.

9. Glutaminase promotes the generation of glutamate, which could be also used for export in exchange of cystine uptake, which is key for ferroptosis inhibition. Thus, the effect of GLS inhibitor on ferroptosis regulation might involve cystine metabolism, which should also be considered and studied.

Response: The reviewer brings up an important point. Glutamate is used by the cellular xCT transporter to bring in cystine, an essential component for the synthesis of glutathione, the cellular defense to oxidative damage. Thus, blockade of GLS activity would be predicted to increase ferroptosis and has been shown to do that in some studies (PMID37969394). Consistent with our results blockade of GLS activity has been shown to rescue peroxide induced cell death (PMID: 37339981). Thus, the impact of GLS activity on ferroptosis appears to be context dependent. We have added this in the text lines 641-647.

10, The study didn't answer the core question of why oHSC could enhance glucose and glutamine metabolism.

Response: A detailed evaluation of the mechanism by which oHSV manipulates cellular metabolic pathways is a very interesting question that we will study in detail for a subsequent report. We believe it is beyond the scope of this current manuscript. Nevertheless, this is the first report to identify oHSV treatment induced metabolic changes in glucose and glutamine flux, that together alter cellular biology to sensitize to lipid peroxidation. The findings have significant implications for the treatment of mutant and wt IDH glioma patients.

11, It's interesting to figure out the exact of ferroptosis in tumor cells and ferroptosis-mediated immune activation respectively in oHSC-induced tumor suppression, in other words, which one is dominant in terms of killing tumor cells, ferroptosis in tumor cells or ferroptosis-mediated immune activation?

Response: Oncolytic viruses induce tumor destruction by two distinct methods: oncolysis, which initiates direct virus induced tumor cell killing and also by immunolysis, which directs immune cell mediated killing of tumors. We believe that an oncolytic burst is important to release tumor antigens and activate and awaken the immune suppressed tumor microenvironment that can lead to destruction of injected and un-injected lesions. The impact of which mechanism is more important for therapeutic benefit likely depends on different tumors and their microenvironment. The reviewers are referred to a detailed review for a more detailed discussion on this subject. PMID: PMC7015832 PMID: 32071927.

REVIEWERS' COMMENTS

Reviewer #1 (Remarks to the Author):

The authors have addressed my critiques/concerns to satisfaction. In my opinion, the manuscript is ready for publication in this journal.

Response: Thank you.

Reviewer #2 (Remarks to the Author):

This revised manuscript by Sahu et al, is highly significant for the field of oncolytic virotherapy and neuro-oncology. The results are timely, rigorous, and the conclusions are in line with the data presented by the authors.

Please find enclosed below minor comments:

1- In the materials and methods section, the authors should cite the paper by Nunez F. J. et al Sci Transl Med. 2019 Feb 13;11(479):eaaq1427. doi: 10.1126/scitranslmed.aaq1427. PMID: 30760578

Response: Thank you for the suggestion. We have added the above reference in the Methods section: line 435.

2- If available, they should also add the relevant reference for the human cells utilized.

Response: We have added a reference in the Methods section: line 426.

3- The authors should also add a sentence related to the fact that it would be valuable to uncover the mechanism by which oHSV manipulates cellular metabolic pathways, as one of the limitations of their study. Understandably. I do agree with the authors, that this is beyond the scope of the current study.

Response: We have now added a sentence in line 419-421: "In future, a more detailed exploration of the mechanisms by which oHSV manipulates cellular metabolic pathways might provide targets to improve therapeutic outcome."

Reviewer #3 (Remarks to the Author):

Several ferroptosis-related markers, such as mitochondria condensation, lipid peroxidation, and alteration of several proteins, are involved in ferroptosis regulation. However, none of them exclusively indicate the occurrence of ferroptosis, simply because at these stages cells might not be undergoing death yet. The reliable way to demonstrate ferroptotic cell death is the results of cell death suppression by ferroptosis inhibitors.

Response: We understand the reviewer's concern. We have now rephrased the conclusion to figure 5 (Line 217-222) as:

“Collectively our data shows that virus treatment increases glucose flux into the TCA cycle which increases cellular ROS. At the same time oHSV treatment shifts glutamine utilization towards reductive carboxylation to induce fatty acid synthesis. PKC activation induces enzymes such as ACSL4. TEM microscopy further shows smaller mitochondria going through fission with evidence of membrane damage, all considered to be hallmarks of ferroptosis.